# Metamorphism in TDP-43 prion-like domain determines chaperone recognition

Jaime Carrasco [1], Rosa Antón[1], Alejandro Valbuena [2], David Pantoja-Uceda [1], Mayur Mukhi [3], Rubén Hervás[3], Douglas V. Laurents [1], María Gasset [1] & Javier Oroz [1] ✉

The RNA binding protein TDP-43 forms cytoplasmic inclusions via its C-terminal prion-like domain in several neurodegenerative diseases. Aberrant TDP-43 aggregation arises upon phase de-mixing and transitions from liquid to solid states, following still unknown structural conversions which are primed by oxidative stress and chaperone inhibition. Despite the well-established protective roles for molecular chaperones against protein aggregation pathologies, knowledge on the determinants of chaperone recognition in disease-related prions is scarce. Here we show that chaperones and co-chaperones primarily recognize the structured elements in TDP-43´s prion-like domain. Significantly, while HSP70 and HSP90 chaperones promote TDP-43 phase separation, co-chaperones from the three classes of the large human HSP40 family (namely DNAJA2, DNAJB1, DNAJB4 and DNAJC7) show strikingly different effects on TDP-43 de-mixing. Dismantling of the second helical element in TDP-43 prion-like domain by methionine sulfoxidation impacts phase separation and amyloid formation, abrogates chaperone recognition and alters phosphorylation by casein kinase-1δ. Our results show that metamorphism in the post-translationally modified TDP-43 prion-like domain encodes determinants that command mechanisms with major relevance in disease.

Prion-like spreading of aggregated TDP-43 structures plays key roles in the development of amyotrophic lateral sclerosis (ALS), frontotemporal lobar degeneration (FTLD), and other major neurodegenerative diseases[1–6]. Despite being a vital nuclear DNA and RNA binding protein regulating RNA transcription, splicing, transport and translation[7], modified TDP-43 forms cytoplasmic pathological inclusions of peculiar morphology under disease[5,8]. TDP-43 aggregation is primed by its C-terminal low-complexity domain, whose prion-like behavior determines proper protein function and the assembly of stress granules by liquid–liquid phase separation (LLPS) in response to cellular stress[9–12]. TDP-43 aggregation arises upon phase de-mixing and transitions from liquid to solid states, which may lead to cell death[9,13–16]. Although the nuclear bodies formed by TDP-43 following

reversible LLPS are cytoprotective[17], alternative TDP-43 assemblies formed in the cytoplasm during or in parallel to the aggregation process may be harmful[18–20]. Interestingly, whereas the presence of TDP-43 inclusions is often used to define the histopathologic patterns of TDP-43 pathologies[21], increasing evidence indicates that cytotoxicity is caused by aberrancies in phase de-mixing rather than aggregation[13,20,22]. Indeed, TDP-43 aggregation is viewed as a way to protect cells, by titrating the protein away from noxious liquid-like phases[13].

For TDP-43, LLPS is determined by a double α-helix signature populated in the C-terminal, prion-like domain (PLD)[23]. This particular region, covering the segment 321–343 of TDP-43[24], evolves into cross β-sheet structures upon transitions to solid states[24,25], forming

[1]Instituto de Química Física Rocasolano (IQFR), CSIC, E-28006 Madrid, Spain. [2]Centro de Biología Molecular "Severo Ochoa", Universidad Autónoma de Madrid, Cantoblanco, E-28049 Madrid, Spain. [3]School of Biomedical Sciences, University of Hong Kong, Pokfulam, Hong Kong. ✉e-mail: joroz@iqfr.csic.es

the core of fibrillar amyloid structures assembled both in-vitro[26] and in-vivo[8]. Notwithstanding, TDP-43 present in pathological inclusions is profusely modified[5,27] and knowledge on the impact of modifications on the harmful phase separation of TDP-43 is still scarce[28–30]. Of the distinct post-translational modifications, methionine sulfoxidation (MetO), which transforms methionine sulfur atoms into the more hydrophilic and voluminous (R, S)-sulfoxide groups, accumulates under oxidative stress due to redox homeostasis impairment and constitutes a covalent signature in prions[31,32]. MetO, a consistent modification occurring in the PLD of accumulated TDP-43 in ALS[27], dissolves preformed condensates[29] and its occurrence is sterically and polarly incompatible with the available TDP-43 amyloid structures[8,26]. However, despite its relevance in disease, the structural impact of this post-translational modification on TDP-43 PLD remains undisclosed.

Since biomolecular condensation is a prolific process in cells under stress, questions arise regarding the factors that separate functional from aberrant phase separation[18]. Molecular chaperones are responsible for triaging proteins into phase-separated condensates under stress[33], coexist with TDP-43 in anisotropic distributions inside nuclear condensates[30] and define the transitions to gel-like assemblies both in nuclear[30,34] and cytosolic TDP-43 condensates[35]. Still, failure to manage increased protein misfolding within the confined limits of phase-separated condensates triggers the conversion of functional granules into pathogenic entities[20]. Recent structural advances in chaperone complexes have shown that structured elements, even when low-populated or transient, determine chaperone recognition in intrinsically disordered proteins, such as TDP-43 PLD[36]. Therefore, the acquisition of alternative polymorphic structures in prions may disrupt processes sustaining proteostasis.

Considering that the transitions to harmful polymorphs in TDP-43 are determined by mutations, oxidative stress, and proteasome and chaperone inhibition[30,35,37], we reasoned that modifications that ensue in pathology direct the formation of alternative TDP-43 PLD assemblies and interactomes. Here, our description of the novel structures acquired by post-translationally modified TDP-43 PLD by Nuclear Magnetic Resonance (NMR) spectroscopy provides insights into the structural basis of harmful phase separation and its interplay with molecular chaperones and co-chaperones, with impact in pathology.

## Results

### Methionine sulfoxidation impairs the association behavior of TDP-43 PLD

Besides well-conserved structured domains responsible for oligomerization, nuclear export, and RNA binding[7], TDP-43 contains a disordered region at the C-terminus with prion-like properties (Fig. 1A)[12]. This PLD, which covers the region 274–414, contains most of the ALS-associated mutations[7], is rich in methionine residues and is sufficient to bolster the aggregation of full-length TDP-43 in disease[9]. The PLD is able to promote condensates by LLPS in vitro[23] and in cells[38], although with different aging properties relative to those condensates formed by full-length TDP-43[35]. To validate LLPS by the PLD, we monitored protein de-mixing by turbidimetry and optical microscopy (Fig. 1B, C, Supplementary Fig. 1A). Turbidimetry showed that both TDP-43 and its PLD exhibited minimal phase separation at low concentrations (10–20 µM, Fig. 1B, Supplementary Fig. 1C). At higher concentrations, PLD phase separated with a strong dependence on the type and concentration of salt. In the presence of sodium ions, LLPS was significantly more effective, even at lower PLD concentrations (Fig. 1B, C). Fluorescence recovery after photobleaching (FRAP) experiments revealed significant fluidity inside the condensates formed by PLD de-mixing over a wide range of incubation times (Supplementary Fig. 2). To determine the impact of disease-relevant post-translational modifications in PLD de-mixing, the PLD was incubated with $H_2O_2$ to achieve methionine sulfoxidation (termed MetO PLD). Remarkably,

even in conditions that potently favor de-mixing (namely high protein and salt concentrations), MetO PLD displays an impaired phase separation (Fig. 1D, Supplementary Fig. 1B–F), in agreement with previous observations on preformed condensates[29].

NMR spectroscopy was employed to determine the structural output of methionine sulfoxidation on the PLD. While the NMR spectra of 300 µM PLD showed a highly disordered protein and contained signs of protein condensation (Supplementary Figs. 1D, 3)[39], MetO causes significant changes in the NMR spectra (Fig. 1E, Supplementary Fig. 4). Crosspeaks belonging to methionine and nearby residues shift significantly (Supplementary Fig. 5A, B), indicating a strong change in their chemical environment due to oxidation. Assignment of the novel resonances showed that PLD Met residues are fully oxidized following treatment with $H_2O_2$, which is validated by the characteristic shifts in their Cβ resonances (Supplementary Fig. 5B). Remarkably, no oxidation of Trp/Phe residues, which are key stabilizing TDP-43 LLPS[40], was detected by NMR or mass spectrometry (Supplementary Fig. 5C, Supplementary Table 1). NMR signal broadening, which is indicative of exchange processes in the NMR timescale in addition to intermolecular associations, shows that there is a strong broadening in the region including residues 305–345 of unmodified PLD upon phase separation (Fig. 1F). This observation indicates that, upon increasing protein concentrations, PLD self-association is mediated by this hydrophobic region. In addition, crosspeaks belonging to this particular region are absent in the CON NMR spectra acquired for the PLD under LLPS conditions (Supplementary Fig. 6). Since CON spectra are less affected by chemical exchange yet sensitive to fast transverse relaxation, this strongly suggests that the 305–345 region of the PLD is involved in intermolecular interactions in the de-mixed phase. Comparison of the NMR signal broadening revealed that MetO PLD is less prone to self-assembly (Fig. 1F), confirming its decreased de-mixing (Fig. 1D, Supplementary Fig. 1)[29]. $^{15}$N spin relaxation parameters corroborated that the PLD contains rigid elements in its structure[23] and that it is assembled in larger species, while MetO PLD remains highly dynamic and dispersed even at high concentrations (Supplementary Fig. 7, Supplementary Table 2). In addition, $^{15}$N Carr-Purcell-Meiboom-Gill (CPMG)-based relaxation dispersion experiments[41] confirmed that the PLD experienced a conformational exchange in the µs−ms chemical shift timescale, indicative of its assembly into dynamic ensembles[23] (Supplementary Fig. 7D, Supplementary Table 2). Remarkably, lack of exchange for MetO PLD confirms that methionine sulfoxidation abrogates phase separation.

### Methionine sulfoxidation promotes significant disorder

Because structured elements in TDP-43 PLD are critical for phase separation[23], we aimed to monitor any possible conformational transition upon protein de-mixing. The structural tendencies observed in PLD do not vary with increasing protein concentrations (Supplementary Fig. 8)[23], regardless of the phase composition (Fig. 1G, Supplementary Fig. 1). In brief, there is a double α-helix formed between residues 321-331 and 334-343, and there are small tendencies to adopt β-strand structures in the regions 368–381 and 391–399 of the PLD. The additional β-strand expected to form in the low-complexity aromatic-rich kinked segment (LARKS, region 312–317)[13] is not apparent in our conditions. Therefore, the helical segments in PLD, which are present in vivo[13], appear equally populated in dispersed samples, and no additional conformational transitions are observed upon phase de-mixing.

Upon oxidation, the α-helical content is significantly decreased, as shown by NMR secondary chemical shifts (Fig. 1G) and circular dichroism (CD) analysis (Supplementary Fig. 9). The structural data indicate that the α-helices between residues 321-331 and 334-343 are largely dismantled in MetO PLD, while the β-strand structures in the regions 368-381 and 391-399 remain intact (Fig. 1G). These structural effects correlate with the location of Met residues and the chemical

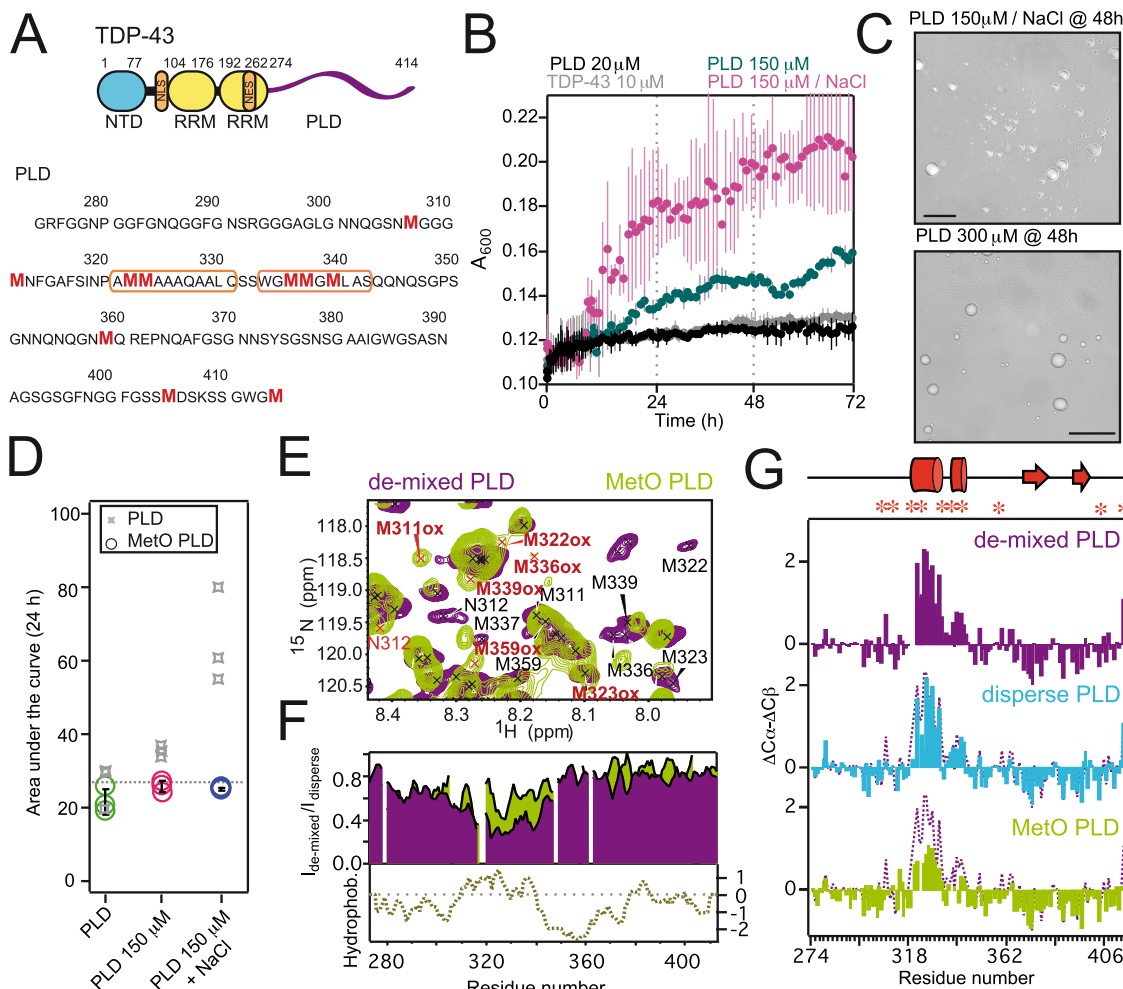

**Fig. 1 | Methionine sulfoxidation impairs TDP-43's PLD phase separation by reshaping its structure. A** Cartoon representation of the domain architecture of TDP-43 (top). The disordered PLD (region 274–414) is represented by a purple line. PLD primary structure, with the Met residues highlighted in red (bottom). The double α-helices are limited by orange boxes. **B** Turbidity measurements showing concentration- and salt-dependent LLPS for the PLD. **C** Differential Interference Contrast (DIC) microscopy images showing liquid condensates formed by the PLD in the specified conditions. Scale bars correspond to 25 μM. **D** LLPS of MetO PLD in the corresponding samples (circles), compared to unmodified PLD (gray stars), as measured by the area under the turbidity curves after 24 h of incubation. Gray broken line corresponds to the averaged LLPS of unmodified PLD. **E** Detailed region of the overlay of the $^{15}$N-HSQC spectra from de-mixed PLD (purple) and MetO PLD (green) showing the large shifts for the Met moieties upon methionine sulfoxidation. MetO cross peaks are highlighted in red. **F** Comparison of the NMR signal intensity of unmodified PLD (purple) and MetO PLD (green) upon de-mixing shows a reduced broadening in the region 305-345 for MetO PLD. The preceding region (280–305) is also partially broadened in both proteins. The plot at the bottom shows the hydrophobicity of the PLD. **G** Secondary chemical shifts ($\Delta C\alpha$-$\Delta C\beta$) analysis for 300 μM PLD (top), 25 μM PLD (middle), and 300 μM MetO PLD (bottom). In these plots, positive values indicate acquisition of α-helical conformations, while negative values correspond to β-strand structures. Each plot shows the overlay with the structural propensities at 300 μM (broken purple line) for comparison. The schematic cartoon at the top highlights the two α-helices (in cylinders) and β-strands (arrows) formed in the PLD. Met residues are located with asterisks. For clarity, Met residues were removed from the MetO PLD plot (bottom) due to the strong shifts upon oxidation (Supplementary Fig. 5B). Unless otherwise stated, turbidimetry and microscopy samples (**B**–**D**) contained 150 mM KCl. NaCl in **B**–**D** refers to 150 mM NaCl. NMR samples (**E**–**G**) contained 10 mM KCl.

shifts observed in MetO PLD (Supplementary Figs. 5A, B) and with the absence of rigid elements observed in MetO PLD $^{15}$N relaxation profiles (Supplementary Fig. 7). Considering the key role of the α-helices in PLD de-mixing[23], the conformational rearrangements help to explain the inability of MetO PLD to phase separate (Fig. 1D, Supplementary Fig. 1). Since fibril formation can be viewed as a liquid-to-solid transition in aging condensates[14–16,18,34,35,42], we next wondered whether the meta-morphism displayed by MetO PLD affects aggregation.

## Oxidized PLD assembles into ultrastructurally different fibrillar aggregates

PLD amyloid aggregates are sustained by a cross-β sheet skeleton in which the side chains of several Met residues are closely packed[8,26]. Conversely, those Met residues lie on the solvent-exposed surfaces of the α-helical segments in solution[43]. Therefore, an increase in polarity and volume of methionine residues side-chains may cause a significant impact on PLD fibril formation. Full methionine sulfoxidation drastically delays PLD fibrillar aggregation as monitored by Thioflavin T (ThT) fluorescence, irrespective of the salt composition (Fig. 2A). Kinetic analysis indicates that the aggregation of PLD and MetO PLD is governed by distinct mechanisms (Supplementary Fig. 10, Supplementary Table 3). In particular, lack of lag phase and of a concentration effect on the $t_{1/2}$ in PLD aggregation suggests that the limiting step is a conformational transition towards amyloid-compatible structures[44], while methionine sulfoxidation severely impairs primary nucleation[45]. This retard in fibril formation kinetics upon methionine sulfoxidation was reproduced for disease-related PLD mutants[46,47], in spite of their intrinsic propensity to form fibrils and expands the kinetic range reported previously[29] (Supplementary Fig. 11). In agreement with ThT measurements, transmission electron microscopy (TEM) showed that

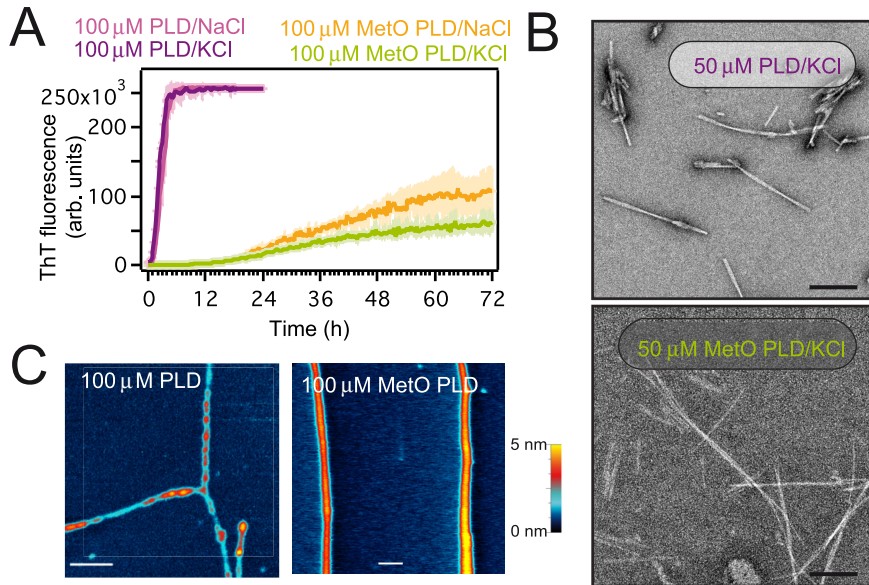

**Fig. 2 | MetO PLD forms distinct fibrillar aggregates. A** Amyloid aggregation kinetics as measured by ThT fluorescence. Samples contained either 150 mM KCl or 150 mM NaCl, as noted. **B** Representative electron micrographs showing fibrils formed by 50 µM PLD and MetO PLD aged for 20 days. Scale bars correspond to 200 nm. **C** Atomic force microscopy topographic characterization of the fibrils formed by 100 µM PLD (left) and 100 µM MetO PLD (right) samples aged for 4 months. PLD fibrils show variable diameters along their length. Scale bars correspond to 400 nm. Heat-color height scale is 5 nm. Samples in **B** and **C** contained 150 mM KCl.

methionine sulfoxidation on the PLD monomer hinders the rate of fibril formation at physiological pH (Supplementary Fig. 12). In addition, fibrils formed by MetO PLD appeared morphologically different from those formed by unmodified PLD at the tested time points (Fig. 2B, Supplementary Fig. 12). Indeed, atomic force microscopy (AFM) shows that the aged amyloid fibrils formed by PLD are unbranched but densely entangled (Fig. 2C, Supplementary Fig. 13), which resemble the TDP-43 inclusions observed in several neurodegenerative diseases[48]. On the contrary, the aged fibrils formed by MetO PLD appear as significantly long, homogeneous, and isolated assemblies (Fig. 2C, Supplementary Figs. 13, 14). These ultrastructural differences may entail the presence of alternative packing arrangements due to methionine sulfoxidation within the fibril core[8,26]. All in all, methionine sulfoxidation causes a significant structural conversion in the PLD which suppresses phase separation and promotes self-association into seemingly alternative aggregates.

### Methionine sulfoxidation fully dismantles the second α-helix

Since MetO structural differences were clustered in the region including residues 312-346 of the PLD (Supplementary Fig. 5A), we used the segment 309-350 (termed $PLD_{309}$) to determine the structure and relaxation properties of MetO PLD by NMR. Analysis of heteronuclear ($^1$H)-$^{15}$N nuclear Overhauser effect (NOE) and $^{15}$N longitudinal ($R_1$) and transverse ($R_2$) rate constants showed that MetO $PLD_{309}$ is highly dynamic in solution (Fig. 3A). Only a minimal increase in the relaxation parameters in the region 327-334 suggests rigidity due to the formation of ordered structures. While this lack of apparent rigidity was corroborated by MetO PLD $^{15}$N relaxation parameters (Supplementary Fig. 7), it is in stark contrast with the backbone picosecond-nanosecond motions of unmodified PLD, which showed local rigidity in this particular region due to the formation of the double α-helix[23] (Supplementary Fig. 7). NMR chemical shifts are also consistent with an overall high structural flexibility, particularly evident in the region where the second α-helix would be located (residues 334-343) (Fig. 3B, Supplementary Fig. 15). In addition, J-coupling analysis showed lower values for the segment 322-331, indicating a tendency to acquire α-helical conformations in this

region of MetO $PLD_{309}$ (Fig. 3C). The rest of the protein showed $^3$J values clustering around 6 Hz, representative of a disordered conformational ensemble[49].

The increased flexibility of MetO $PLD_{309}$ precluded the observation of abundant $^1$H-$^1$H NOE contacts that would define distance restraints for the structure calculation. Nonetheless, using chemical shifts and torsion angle restraints (Supplementary Table 4) a set of 100 conformers were calculated by CYANA[50]. The 20 conformers with the lowest target function were selected and assessed by PROCHECKNMR (Fig. 3D). Direct comparison with the available solution structures of the 311-360 fragment from the PLD (PDB code 2n3x)[43] highlights the absence of the second α-helix due to the clustering of MetO336, MetO337, and MetO339. In addition, MetO322 and MetO323 cause the partial dismantling of the N-terminal part of the first α-helix. The high intrinsic dynamics of the short α-helix between residues Ala324-Ser332 reduces the structural convergence of the conformers obtained (RMSD to the mean coordinates of 0.41 Å for heavy backbone atoms, Supplementary Table 4). Indeed, considering the empirical Cα chemical shifts for 100% α-helical conformers (+3.1 ppm)[51], the short α-helix acquired by MetO $PLD_{309}$ only populates about 27% of the conformational ensemble (Fig. 3B).

### Chaperones recognize PLD but not MetO PLD

The decline of protein misfolding management by molecular chaperones allows the manifestation of various protein aggregation diseases[52]. HSP70 and HSP90 are the main cytoplasmic chaperones, and their refolding activities are tightly regulated by a large cohort of co-chaperones[36]. HSP70, HSP90, and certain co-chaperones prevent the cytoplasmic aggregation of TDP-43 PLD[53-55]. In particular, HSP70 is able to phase separate in combination with TDP-43[30,34], and its proper ATP-dependent activity may determine aberrant phase separation in stress[20]. Because the levels of the inducible isoforms of HSP70 and HSP90 (HSP72 and HSP90α and HSP90β, respectively) substantially increase in stress[56,57], we analyzed their impact on PLD phase separation. All HSP70 and HSP90 isoforms promoted the de-mixing of PLD in a highly similar manner, in a way which resembles the LLPS of highly-concentrated PLD (Fig. 4A, B, Supplementary Fig. 16A, B).

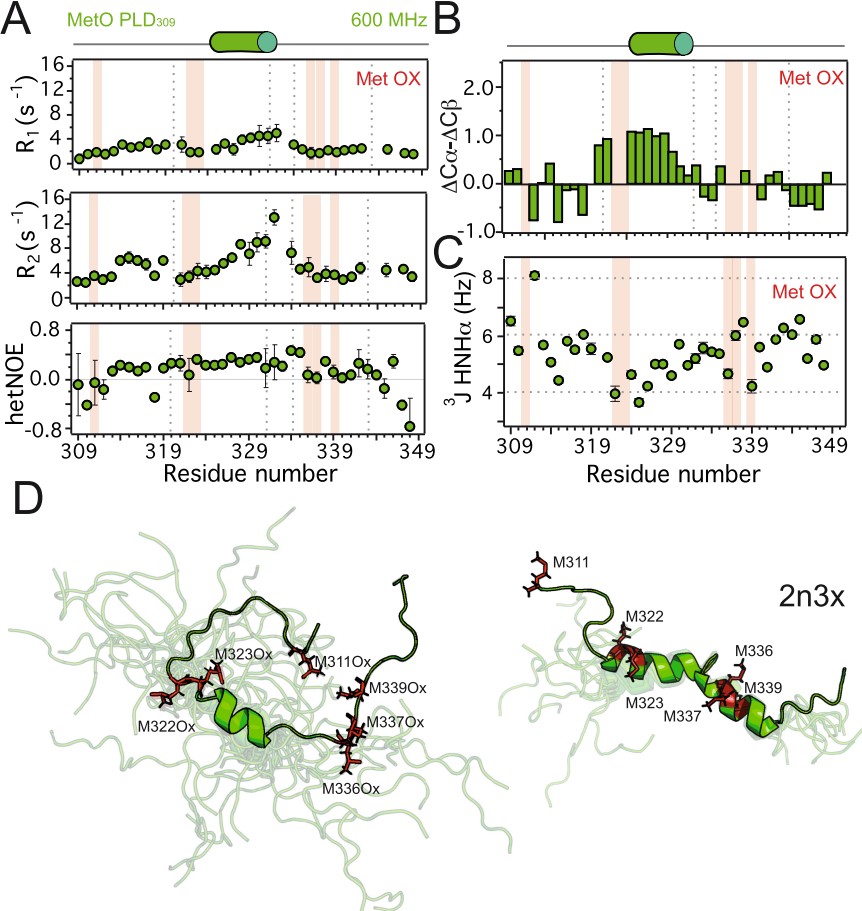

**Fig. 3 | Methionine sulfoxidation promotes significant disorder in the PLD.**
**A** Longitudinal ($R_1$, top), transverse ($R_2$, middle) rate constants, and heteronuclear ($^1$H)-$^{15}$N NOE (bottom) relaxation parameters were obtained for MetO PLD$_{309}$ at 600 MHz. The values of all relaxation parameters indicate that the protein is highly dynamic. **B** Secondary chemical shifts ($\Delta C\alpha$-$\Delta C\beta$) for MetO PLD$_{309}$. **C** J-coupling assessment of the secondary structure propensities for MetO PLD$_{309}$. Values around 4 Hz indicate $\alpha$-helix formation, while values around 8 Hz are typical of $\beta$-strand structures. **D** Structural ensemble of the 20 conformers of MetO PLD$_{309}$ with the lowest conformational energy as calculated by CYANA (left), compared to the ensemble of ten conformers of a comparable PLD fragment (PDB code 2n3x, right). The lowest energy structure is displayed on top of the corresponding ensemble, with the rest of the conformers shown in transparent representation. Structures were aligned onto the structured elements. Met residues are highlighted in red stick representation. Gray broken lines in **A**, **B** plots indicate the boundaries of the two $\alpha$-helices present in the PLD, while light red bars mark the location of Met residues. Diagrams on top of the plots (**A**, **B**) represent the $\alpha$-helix formed in MetO PLD$_{309}$ (green cylinder).

Residue-by-residue analysis of the interactions established by the PLD and chaperones by NMR spectroscopy showed that chaperones recognized the region 321-343 in the PLD (Fig. 4C). In particular, the four chaperones tested were able to bind in identical fashion to the double α-helical motif in the PLD. Remarkably, the signal broadening observed in dispersed PLD upon chaperone binding resembled very closely that observed in de-mixed PLD, suggesting that chaperones promote intermolecular associations in the PLD via the double α-helical motif. On the contrary, recognition of the PLD by HSP70 and HSP90 chaperones was suppressed by methionine sulfoxidation (Fig. 4D). Since chaperones bind to the structured regions of the PLD (Fig. 4C), impairment of both hydrophobicity and α-helical motifs in MetO PLD (Fig. 3D) dictates the evasion from chaperone recognition and commands its accumulative trend.

### HSP40 co-chaperones show divergent effects on PLD phase separation

Given the relevance of the large HSP40 co-chaperone family on TDP-43 proteinopathies[54,58,59], we characterized PLD phase separation in the presence of representative members of the different J-domain containing (JDP) subfamilies: DNAJA2 (class A), which is a potent tau aggregation inhibitor;[60] DNAJB1 and DNAJB4 (class B), which promote

the clearance of TDP-43 deposits;[54] and DNAJC7 (class C), which is considered an ALS gene[58] (Supplementary Fig. 17). Remarkably, while DNAJA2 and DNAJC7 suppressed PLD de-mixing, both DNAJB1 and DNAJB4 strongly promoted PLD phase separation (Fig. 4E, F, Supplementary Fig. 16C). Despite the synergy with HSP70 presented by DNAJB1 and DNAJB4 promoting PLD phase separation (Fig. 4E), we wondered if the divergent effects shown by JDPs were based on the recognition of different regions of the PLD. Intriguingly, the four HSP40 isoforms selectively bound the double α-helical motif in the PLD in comparable fashion (Fig. 4G).

The PLD:HSP40 interaction profiles remained largely identical upon the addition of HSP70, which suggests that the PLD stays bound to HSP40 in the ternary (PLD:HSP70:HSP40) complex (Fig. 4G). Evidence of interaction in additional PLD regions containing aromatic residues (e.g., F313, F316, F367, Y374, W385, F397, F401) upon addition of HSP70[61] indicates formation of the PLD:HSP40:HSP70 ternary complex, which can be isolated using a chemical crosslinking approach (Supplementary Figure 18). Therefore, the divergent effects observed on PLD phase separation by JDPs are not based on differential structural recognition, but could be related to differences in LLPS propensities shown by the JDPs[59]. In particular, DNAJA2 contains a GF-rich region adjacent to the J domain (residues 70–130 of human

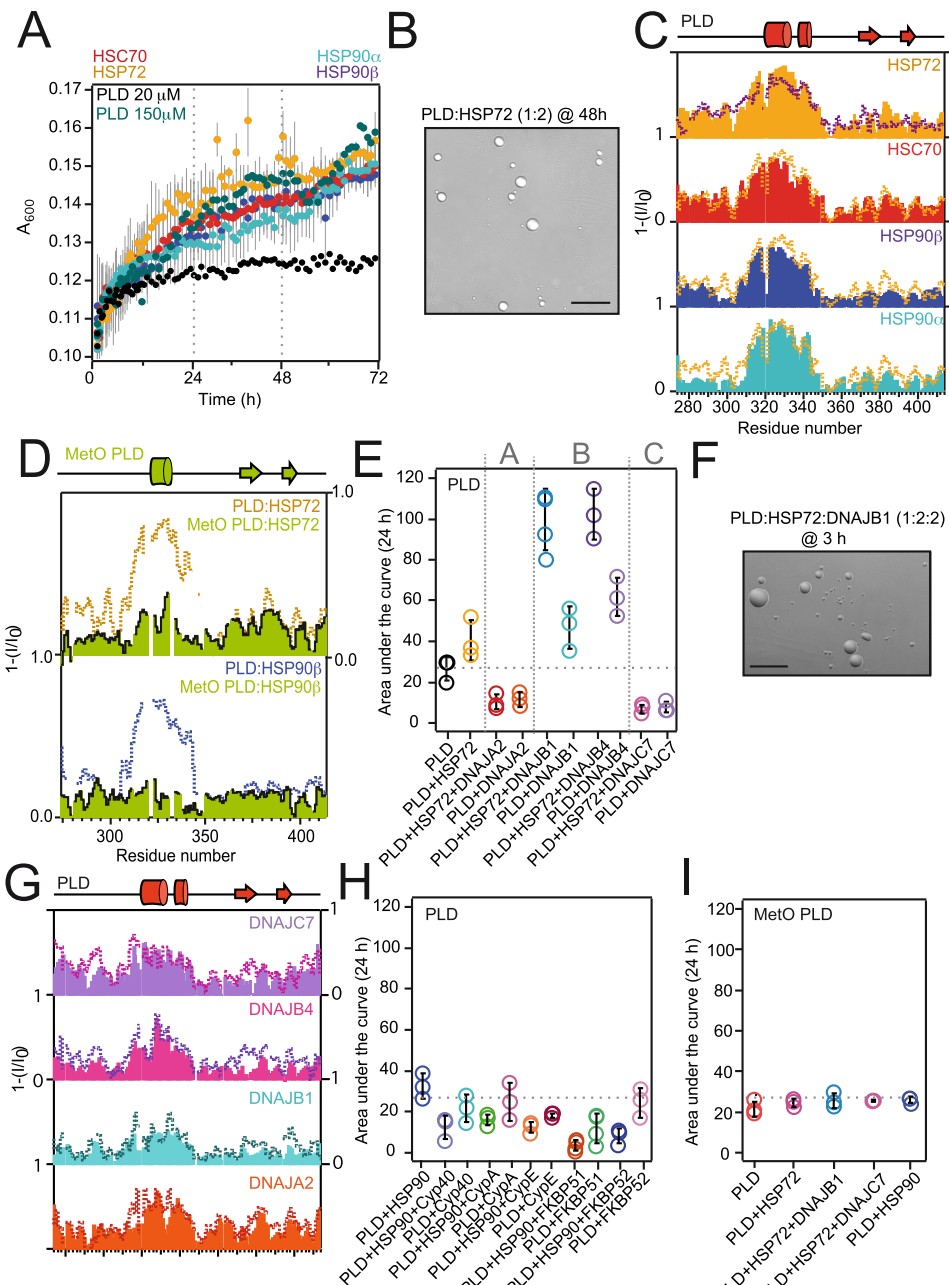

**Fig. 4 | The interplay with chaperones and co-chaperones is mediated by the double α-helix of the PLD. A** Turbidity measurements of the PLD in presence of the indicated chaperones (all in 1:2 molar ratios). Error bars are not included in the PLD plot (black dots) for clarity. **B** DIC microscopy image of the condensates formed by 20 μM PLD in complex with HSP72 after 48 h of incubation at 25 °C. **C** NMR signal intensity plots of 25–35 μM PLD in complex with the indicated chaperones (all in 1:2 molar ratios). For comparison, the plots are overlaid with the data corresponding to de-mixed PLD (300 μM PLD, broken purple line, Fig. 1F) and HSP72 interaction (orange broken line). **D** NMR intensity plots for MetO PLD in the presence of the chaperones (green bars) in comparison to the unmodified PLD:chaperone interactions in identical molar ratios (broken lines). **E** LLPS of 20 μM PLD in the presence of HSP72 and/or JDPs represented as the area under the curve of the turbidity measurements after 24 h. Vertical broken lines separate the three JDP classes (A, B, and C, indicated on top). **F** DIC microscopy image of the condensates formed by 20 μM PLD in complex with HSP72:DNAJB1 after 3 h of incubation at 25 °C. **G** NMR signal intensity plots for the interaction of 25–35 μM PLD with JDPs (all in 1:2 molar ratios). In each plot, the broken line represents the interaction of the PLD with HSP72 and the corresponding JDPs (all in 1:2:2 molar ratios). **H** LLPS of 20 μM PLD in presence of HSP90 and the specified co-chaperones as measured by turbidity. **I** LLPS of 20 μM MetO PLD in the presence of the specified chaperones and co-chaperones. For simplicity, the plots in **C**, **D**, and **G** show the reverse of the NMR signal decay. The gray broken line in **E**, **H**, and **I** represents the average turbidity of 20 μM PLD, for comparison. Scale bars (**B**, **F**) correspond to 25 μM. Turbidimetry and microscopy samples (**A**, **B**, **F**, **H**, **I**) contained 150 mM KCl, whereas NMR samples (**C**, **D**, **G**) contained 10 mM KCl.

DNAJA2, Supplementary Fig. 17) which is rich in cationic residues (pI = 8.5) and has an elevated propensity to phase separate according to FuzDrop[62]. Since the PLD is also positively charged (pI = 10.0), the chances for PLD establishing heterologous assemblies with the GF region of DNAJA2 are minimal. On the contrary, the GF region of

DNAJB1 and DNAJB4 (spanning residues 70–158) also have a high propensity to phase separate but are enriched in anionic residues (pI = 5.5), which would allow electrostatically-driven associations with the PLD in the de-mixed phases. Finally, DNAJC7 contains two disordered regions flanking the J domain with a low overall pI (5.2), but

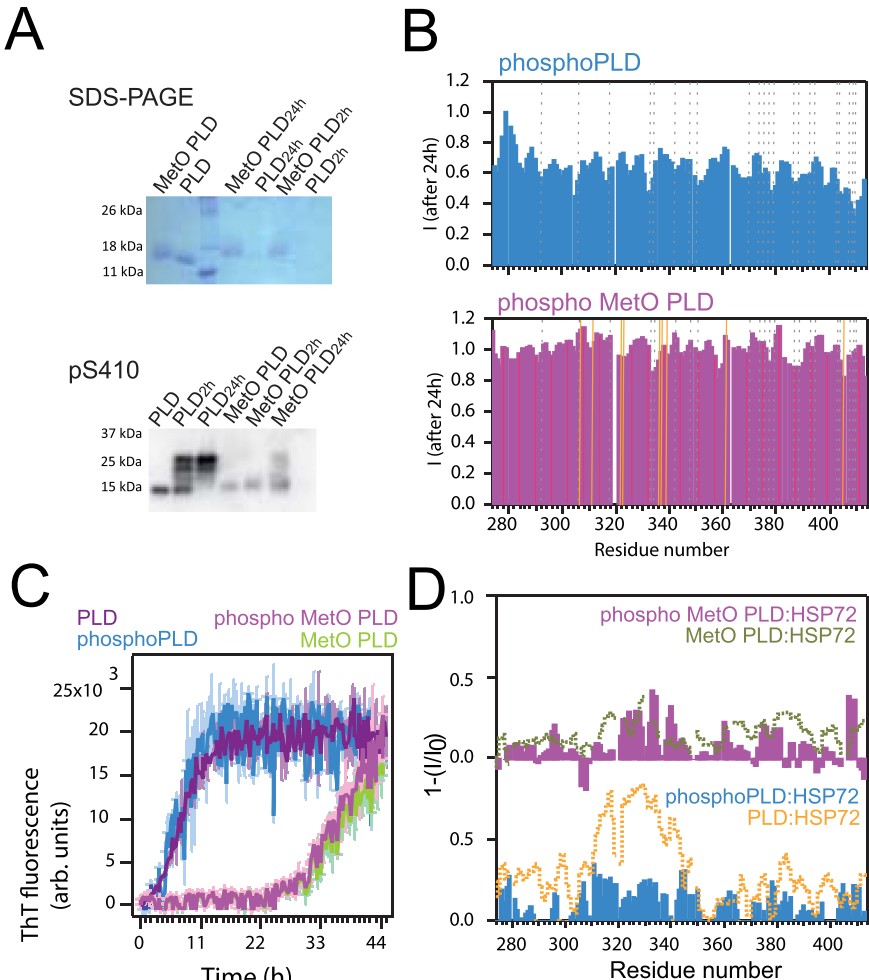

**Fig. 5 | MetO impairs CK1δ phosphorylation. A** Western blot immunoblotting results for the phosphorylation of the PLD using pS410 (bottom) antibody. CK1δ incubation times are indicated. Band at 14 kDa corresponds to unphosphorylated PLD, and phosphorylation is revealed as an increase in the molecular weight. On top, the same samples are subjected to SDS-PAGE, for comparison. Phosphorylated PLD is undetected in the gel due to aggregation. MetO samples are noticeably detained during migration in the gel. **B** NMR signal intensity decay plots after 24 h of incubation at 25 °C for 40 μM phosphoPLD (top) and phospho MetO PLD (bottom). The decay in intensity observed in phosphoPLD is attributed to sample precipitation. Broken gray lines locate Ser resides and golden lines locate Met residues. **C** Kinetics of amyloid fibril formation as measured by ThT fluorescence for 20 μM PLD samples. **D** Reverse of the NMR intensity plots for 40 μM phosphoPLD (bottom, blue) and phospho MetO PLD (top, magenta) in the presence of HSP72 (all in 1:2 molar ratios). For comparison, the plots are overlaid to the data for PLD:HSP72 interaction (bottom, orange line, corresponding to Fig. 4C) and MetO PLD:HSP72 (top, green line, corresponding to Fig. 4D) in identical molar ratios.

which do not show a clear propensity to phase separate according to FuzDrop[62]. Therefore, the divergent effects of HSP40 isoforms on PLD phase separation could be explained by the propensity to establish heterologous de-mixed assemblies, adding yet another layer of complexity to the diverse actions of this co-chaperone family[63,64].

In addition, we also monitored PLD de-mixing in the presence of several HSP90 co-chaperones, which regulate different processes of the HSP90 activation cycle[36]. All the HSP90 co-chaperones tested (namely CypA, CypE, Cyp40, FKBP51, and FKBP52) suppressed PLD phase separation, even in the presence of HSP90 (Fig. 4H). Finally, because chaperones do not recognize the novel polymorphic structures in MetO PLD (Fig. 4D), they remain unable to modulate the de-mixing of MetO PLD, even in the presence of JDPs (Fig. 4I). This evidence is in agreement with the reported inability of oxidized PLD to phase separate[29] (Supplementary Fig. 1). Overall, all the MetO PLD structural features support a novel accumulative pathway in the absence of proper proteostasis control.

### Phosphorylation by CK1δ is altered in MetO PLD

The presence of hyperphosphorylated TDP-43 chains is the pathognomonic feature of the inclusions isolated from ALS and FTLD patients[5]. However, because the core of the fibrillar amyloid aggregates obtained from patients' brains does not show phosphorylation modifications[8], it is unclear whether phosphorylation occurs in the preformed aggregates or in a later stage, as an attempt to eliminate the aggregates[8,65]. Alternatively, phosphorylation may affect TDP-43 balanced phase separation[28], providing a basis for the prevalence of phosphorylation traits in disease and its potential correlation with disease subtypes[6]. Therefore, determining the structural impact of phosphorylation on the PLD and its crosstalk with methionine oxidation will provide relevant insights into the role of modifications in protein diseases.

Casein kinase-1 (CK1δ) has been postulated as one of the kinases involved in the hyperphosphorylation of TDP-43 in vivo[65,66]. Phosphorylation of the PLD by CK1δ was confirmed by immunoblotting with pS410 antibody, which recognizes phosphorylated Ser 410 in the PLD (Fig. 5A, Supplementary Fig. 19)[5]. Interestingly, significant aggregation was evident upon phosphorylation (Fig. 5A, Supplementary Figs. 19, 20). In addition, pS410 antibody revealed that phosphorylation was drastically diminished in soluble MetO PLD, likely because the oxidative modifications interfere with the client identification process by CK1δ[67]. Phosphorylated PLD (termed phosphoPLD) was also

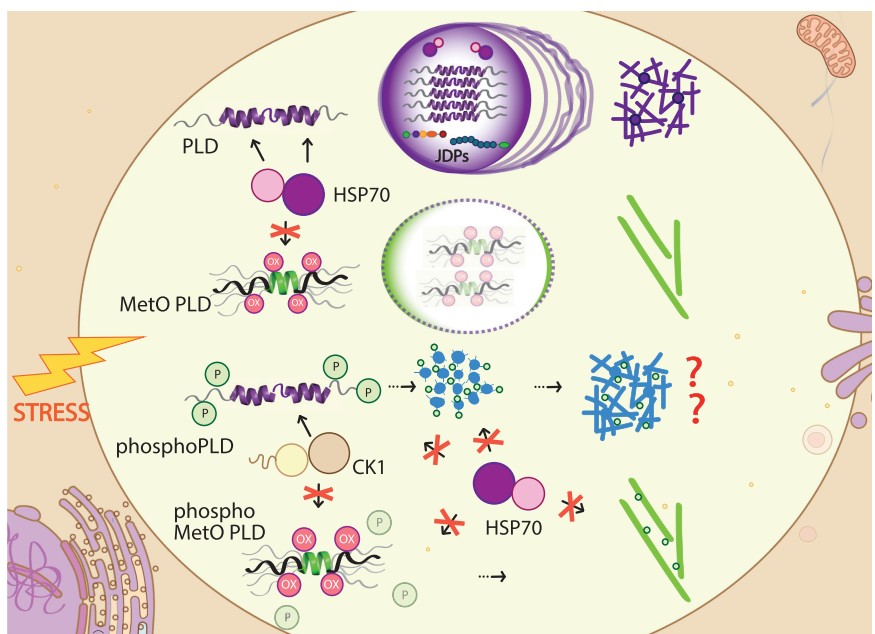

**Fig. 6 | Mechanistic model for the role of the modifications in the PLD in TDP-43 pathogenicity.** TDP-43 PLD phase separation is strictly controlled by HSP70 and JDPs, whose interaction is mediated by structured elements present in the PLD. A liquid-to-solid transition will promote aggregate and fibril formation. Under oxidative stress, methionine sulfoxidation of the PLD will promote structural changes that will abrogate chaperone control and impact PLD phase separation, leading to the formation of alternative mature amyloid fibrils. While CK1δ phosphorylation promotes the aggregation of the PLD and hampers its recognition by HSP70, phosphorylation of soluble PLD is prevented after methionine sulfoxidation. Overall, modifications in the PLD trigger metamorphism which determines chaperone recognition, with impact on TDP-43's pathophysiology.

characterized by NMR spectroscopy, which confirmed the reduced phosphorylation of soluble MetO PLD (Fig. 5B, Supplementary Figs. 19, 20).

Substantial precipitation of the phosphoPLD samples compromised the study of the structural impact of serine phosphorylation by NMR (Fig. 5B). The secondary chemical shifts analysis suggests that the structure of phosphoPLD in solution is comparable to unmodified PLD (Supplementary Fig. 21). This observation could be explained by the limited CK1δ phosphorylation of the serine residues located in between the double α-helix of the PLD (namely Ser332 and Ser333)[43,65]. In addition, it is possible that these results are largely contributed by the residual, non-phosphorylated PLD that remains in solution (Fig. 5A, B). Importantly, phosphorylation does not induce a strong shift in structured populations in MetO PLD (Supplementary Fig. 21), in agreement with its negligible phosphorylation in the soluble state (Supplementary Fig. 20). Moreover, the kinetics of fibrillar aggregation by PLD and MetO PLD assessed by ThT fluorescence was not affected by CK1δ phosphorylation (Fig. 5C). Overall, phosphorylation of the PLD promotes its aggregation but does not affect any secondary nucleation[45], likely because phosphorylation does not act on the α-helical region of the PLD. However, only fibrillar MetO PLD is a convenient substrate for CK1δ (Supplementary Fig. 20).

Interestingly, NMR showed that the recognition of both unmodified and MetO PLD by HSP70 was significantly diminished upon incubation with CK1δ (Fig. 5D). Immunoblotting of the NMR samples revealed that phosphorylation induced precipitation of the PLD, whereas HSP70 remained in the soluble fraction (Supplementary Fig. 19D, E). These intriguing results indicate that the aggregation of the PLD promoted by phosphoPLD titrates the protein away from a potential chaperone binding. Besides, methionine sulfoxidation makes PLD monomer inaccessible for both CK1δ and HSP70 (Fig. 5A, D, Supplementary Fig. 19). Therefore, while acting through different mechanisms, both modifications (methionine sulfoxidation and serine phosphorylation) reshape TDP-43 interactome, which may have a pathological outcome.

## Discussion

The structural basis of TDP-43 de-mixing and aggregation are well described. In brief, the helical region between residues 321-343 of the C-terminal PLD from TDP-43 is responsible for LLPS and chaperone binding[23,34,43]. However, the PLD of the TDP-43 contained in the aggregates formed under neurodegeneration is covalently modified[5,27] endowing different structural and interactome landscapes and accumulative traits. Indeed, the quantitatively sulfoxidized PLD accommodates into highly disordered, polymorphic conformations in which the key double α-helix motif is dismantled. While the first α-helix leads PLD de-mixing[23], the second α-helix plays a role in the aggregation propensity[43]. Consequently, MetO PLD fails to phase separate and function as client for chaperones, co-chaperones, and kinases (Fig. 6). However, it self-aggregates into morphologically distinct fibrils underlining a switch in its interaction capacity. Overall, methionine sulfoxidation is a post-translational modification with significant impact on amyloid aggregation[68]. In this scenario, artificial substitutions of the helical region methionines from the PLD with residues with polar/charged side chains (E/D/S/T/Q/N/K/R) have shown increased cellular toxicity[13]. Additionally, mice containing the TDP-43 M323K variant (which mimics MetO polarity) exhibit a progressive motor phenotype linked to a splicing gain-of-function[46]. Here we show that MetO PLD remains in the disperse phase avoiding proteostasis control, and providing the basis for its potential toxic effect. Accordingly, determining in detail the structural output of modified TDP-43 metamorphs could help unveil the morphological and cytotoxic heterogeneity of TDP-43 aggregates in ALS and FTLD subdisease types[69].

The structural changes induced by MetO on PLD associations appear to play a double role as a function of their reversibility. Oxidative stress is known to promote the recruitment of TDP-43 into stress granules via disulfide linking of the RNA-recognition motifs domains and the aid of PLD condensation[70]. However, if the PLD methionines become sulfoxidized this domain may not phase separate properly, allowing aberrant engagements. In addition, phosphorylation of the PLD has been deemed as a protective intervention to isolate

awry protein forms[71]. Our data show that CK1δ phosphorylates PLD and enhances its aggregation, in addition to the presumed phosphorylation of PLD aggregates[8,65]. On the contrary, MetO PLD becomes phosphorylated by CK1δ when aggregated suggesting an amiss feature in sulfoxidized PLD fibrils. Reduction of the sulfoxide group in MetO is catalyzed by the conserved methionine sulfoxide reductases (MSR), which contain two classes and four isoforms (MSRA, MSRB1, MSRB2, and MSRB3) in mammals[72]. If the function of MSRs were impaired, methionine sulfoxidation could serve as a rheostat to regulate phosphorylation under oxidative stress[67]. Therefore, the specific crosstalk between phosphorylation and methionine oxidation, among other modifications, might be key defining TDP-43's metamorphism at the initial stages of the disease.

TDP-43 noxious cellular effects may be triggered by increased PLD misfolding within condensates which can overpower elements of the proteostasis network such as molecular chaperones for a first clearance attempt[20,33,35]. Molecular chaperones play an important role in TDP-43 clearance in disease[11,30,35,53,54,70]. Indeed, chaperone levels are lower in cells and tissues of ALS patients, which correlate with the abundance of TDP-43 inclusions[54,73]. TDP-43 is constitutively bound to members of the HSP70 and HSP40 families through the PLD[11], and both HSP70 and HSP40 chaperones undergo LLPS[34,59]. Yet, they fail to facilitate MetO PLD phase separation due to an absence of recognition. Overall, understanding how TDP-43 evades or outpaces chaperone control appears crucial to establish the molecular mechanisms triggering its pathogenicity in ALS and other dementias (Fig. 6).

A recent report revealed that HSP70 selectively binds the region 310-345 of the PLD[34]. Our NMR data show that not only HSP70 but also HSP90 chaperones recognize that segment of the PLD. This PLD region is particularly hydrophobic, with a significant content of Ala and Met residues. In particular, the region 310-330, which covers the first α-helix and its preceding segment[43] is enriched in hydrophobic residues, followed by a linker connecting to the second α-helix which is filled with polar residues (fragment 331-334). While HSP70 and HSP90 chaperones bind to the whole 310-345 segment, our NMR data indicate that chaperone recognition seems stronger in the first α-helix and the preceding segment of the PLD (residues 310-330). This region is almost identical to the segment of the PLD recognized by the chaperone HSPB1[35]. Additional hydrophobic stretches of the PLD containing aromatic residues[61] bind weakly to chaperones (residues 380-400). Therefore, hydrophobic interactions seem to be the driving force for PLD binding to chaperones. Similarly, cellular HSP70 and HSP90 recognize a hydrophobic, α-helical segment in α-synuclein[74]. This is in contrast to the promiscuous interaction of tau with HSP90, largely governed by electrostatics and further contributed by the short hydrophobic elements of tau[75]. Increasing structural knowledge on the complexes formed by chaperones and misfolded or disordered clients enables us to identify the client motifs driving recognition in prions[36].

Intriguingly, JDP co-chaperones seem to bind to the PLD in a more selective fashion, at least for the members of the A and B families. Particularly, binding of DNAJA2 to the PLD appears restricted to the double α-helix in the PLD (region 321-343) with the preceding 310-320 segment remaining accessible for HSP70 interaction in the ternary complex. DNAJB4 selects the first α-helix of the PLD, whereas DNAJC7 appears to bind scattered regions of the PLD. Differences in recognition might result from the JDP domains involved. In this sense, only the DNAJB1 CTD domains account for α-synuclein binding[64]. Since the CTD domains of DNAJB1, DNAJA2, and DNAJB4 are highly conserved, it can be argued that PLD recognition by class A and B JDPs also involves their CTD domains. However, since DNAJC7 lacks the CTD domains but retains the conserved J domain and two flanking flexible regions (segment 361-494 from DNAJC7), it is tempting to assign them the recognition capacity. Accordingly, determining whether the J domains and the flexible, LLPS-promoting regions in JDPs participate in PLD recognition would be relevant to ascribe the mechanisms for JDPs

facilitating PLD phase separation (Fig. 6)[59]. The complexity of the chaperone network is somewhat embodied in the large number of naturally occurring JDP isoforms in the human proteome, with different domain architectures and activation mechanisms[63,64]. Indeed, use of different domains may explain the disparities observed among JDPs in TDP-43 clearance efficiency[54]. Considering that the targeted clearance of TDP-43, in particular by co-chaperones, is a key strategy for therapeutic intervention in ALS and FTLD[55], insights into client specificity by JDPs are decisive to nominate potential disease biomarkers[11,58]. Still, here we show that post-translational modifications dramatically alter the structural properties in metamorphic proteins impairing chaperone recognition. It is therefore of paramount importance to include the relevant post-translationally modified chains as the targets for disease intervention.

## Methods

### Sample preparation

Both full-length human TDP-43 (UniProt KB Q13148, [https://www.uniprot.org/uniprotkb/Q13148/entry]) and its fragment 274-414 (PLD) were subcloned in modified pET28a vectors (Novagen) containing Thioredoxin (TXA) as a fusion protein followed by a six-histidine tag for Ni²⁺ affinity purification and a Tobacco Etch Virus (TEV) protease cleavage site. All cloning procedures were performed following the Gibson Assembly method (New England Biolabs). The PLD fragment was used as a template for the generation of M323K and M337V, addition of a C-terminal cysteine residue (C415) for the covalent attachment of a fluorescent tag (detailed below), and cloning the fragment 309-350 of TDP-43 (PLD$_{309}$) using the same plasmid platform. Human HSC70, HSP72, HSP90α, HSP90β, DNAJA2, DNAJB1, DNAJB4, DNAJC7, Cyp40, CypA, CypE, FKBP51, and FKBP52 cDNA sequences were cloned into respective pET28a vectors (GenScript). All sequences were verified by DNA sequencing. Cloning procedures were performed with the following oligonucleotides in both DH5α and XL1 *E. coli* strains:

| | Forward | Reverse |
|---|---|---|
| **TDP-43 insert** | 5′ATTTCCAGGGATCCATGTCTGAATATATTCG3′ | 5′GTGGTGCTCGAGTTACATTCCCCAGCCAGAAG3′ |
| **TDP-43 vector** | 5′CTCGAGCACCACCACCACCAC3′ | 5′GACATGGATCCCTGGAAATACAGGTTTTC3′ |
| **PLD insert** | 5′ATTTCCAGGGATCCGGAAGATTTGGTGGTAATC3′ | 5′GTGGTGCTCGAGTTACATTCCCCAGCCAGAAG3′ |
| **PLD vector** | 5′CTCGAGCACCACCACCACCAC3′ | 5′CTTCCGGATCCCTGGAAATACAGGTTTTC3′ |
| **PLD$_{309}$ vector** | 5′CTCGAGCACCACCACCACCAC3′ | 5′ GAAAACCTGTATTTCCAGGGATCCGGTGG3′ |
| **PLD$_{309}$ insert** | 5′ ATTTCCAGGGATCCGGTGGGATGAACTTTG 3′ | 5′ CAGTCAGGCCCATCGTAACTCGAGCACC 3′ |
| **M323K** | 5′AATCCAGCCATGAAGGCTGCCGCC3′ | 5′ GGCGGCAGCCTTCATGGCTGGATT3′ |
| **M337V** | 5′CAGAGCAGTTGGGGTATGGTGGGCATGTTAGCC3′ | 5′GGCTAACATGCCCACCATACCCCAACTGCTCTG3′ |
| **PLD C415** | 5′CTTCTGGCTGGGGAATGTGTTAACTCGAGC3′ | 5′GTGGTGCTCGAGTTAACACATTCCCC3′ |

TDP-43, PLD (including WT and its variants) and PLD$_{309}$ fragments were produced in Rosetta 2 (DE3) *E. coli* strain, inducing with 0.5 mM isopropyl β-d-1-thiogalactopyranoside (IPTG) at OD$_{600}$ = 0.7 at 25 °C for 20 h. For ¹⁵N/¹³C isotopic labeling, M9 minimal media was used including 4 g/l of ¹³C-glucose and 1 g/l of ¹⁵NH₄Cl as the sole sources of C and N, respectively. Cells were harvested by 20-min centrifugation at 5300 *g*, resuspended in 30 ml/L of lysis buffer (20 mM Tris-HCl/500 mM NaCl/10 mM imidazole [pH 8]), including 5 µl of protease inhibitors (ThermoFisher Scientific), 0.08 mg/ml of DNAse I and RNAse

A (both from Sigma-Aldrich) and 1 mg/ml lysozyme (Sigma-Aldrich) and sonicated. After centrifugation at 15,500 $g$ for 20 min, the pellet was resuspended in the aforementioned buffer, incubated for 15 min at 37 °C and harvested by centrifugation. Pellets were dissolved in 20 mM Tris-HCl/500 mM NaCl/8 M urea/10 mM imidazole [pH 8], clarified by centrifugation at 15,500 $g$ during 20 min and loaded onto pre-equilibrated Ni$^{2+}$ affinity columns (ABT), adding 150 mM imidazole to the buffer for the elution. TEV digestion was performed in 20 mM Tris-HCl/150 mM NaCl/1.5 M urea [pH 8] buffer (TEV buffer), by overnight incubation at 25 °C with 0.5–1 mg/ml of TEV protein (produced in house) per 15 mg of fusion protein. Cleaved TDP-43 and PLD proteins were directly cleared with a Ni$^{2+}$ affinity purification as previously detailed. Buffers for the purification and cleavage of the PLD C415 variant were supplemented with 10 mM DTT. After purification, protein samples were concentrated using an ultrafiltration Merck Amicon™ bioseparation stirring cell with 10 kDa filter with a continuous Argon flow. Protein samples were snap-frozen at −80 °C in small aliquots at concentrations around 200 µM. For PLD$_{309}$, cleaved proteins were purified by Ni$^{2+}$ affinity in the absence of urea, dialyzed into NMR buffer (described below), lyophilized, and stored at −20 °C.

Thawed TDP-43 and PLD aliquots were clarified by centrifugation at 15,000 $g$ for 15 min, followed by filtration through a 100 kDa filter (Microcon) at 10,000 $g$ to remove preformed aggregates. Buffer exchange into 20 mM Hepes/10 mM KCl/0.03% sodium azide [pH 6.8] (LIF buffer) and 20 mM Hepes/10 mM KCl/5 mM MgCl$_2$/1 mM ATP/1 mM DTT/0.2 mM phenylmethylsulfonyl fluoride (PMSF)/0.03% sodium azide [pH 6.8] (NMR buffer) was performed using Zeba desalting columns (ThermoFisher), followed by additional clarification steps at 15,000 $g$ to remove aggregates. If required, a concentration step using Amicon concentrators with 10 kDa and 3 kDa cutoff filters (Millipore) was performed. Protein concentration was determined spectrophotometrically using the protein's molar extinction coefficients (44,920 and 17,990 cm$^{-1}$ M$^{-1}$ for TDP-43 and PLD, respectively).

Molecular chaperones and co-chaperones were produced in Rosetta 2 (DE3) and BL21 (DE3) star *E. coli* strains, using 1 mM IPTG for induction at OD$_{600}$ = 0.8–0.9 during 4 h at 37 °C or 16 h at 25 °C. Cells were harvested and lysed as explained previously. Recombinant proteins in the soluble fraction were purified by Ni$^{2+}$ affinity chromatography using high-density NiSO$_4$ agarose beads (ABT), 20 mM Tris-HCl/500 mM NaCl/10 mM imidazole [pH 8] as binding buffer, and 500 mM imidazole for elution. Proteins were further purified by size exclusion chromatography using HiLoad™ 26/60 Superdex™ 75 pg columns (Cytiva) in 10 mM Hepes/500 mM KCl/5 mM DTT [pH 7.5]. Fractions containing monomeric protein were pooled, concentrated using Vivaspin® 20 ultrafiltration tubes (Sartorius) and stored at −80 °C. Chaperone and co-chaperone samples were buffer exchanged before the experiments into the abovementioned buffers using Zeba spin desalting columns (ThermoFisher).

### PLD covalent modifications

To achieve methionine sulfoxidation (MetO), aliquots of PLD and PLD$_{309}$ were thawed or resuspended (for PLD$_{309}$) in the corresponding buffer and clarified at 15,000 $g$ for 15 min prior to their incubation with 10 mM H$_2$O$_2$ at 25 °C for 16 h. Full MetO was validated by mass spectrometry and NMR spectroscopy. Excess H$_2$O$_2$ was removed by buffer exchange before the experiments using Zeba spin desalting columns (ThermoFisher).

Phosphorylation was assayed by incubating 20–40 µM protein samples at 37 °C with 0.4 µM CK1δ (CK1δ residues 1-294, Sigma-Aldrich) in 20 mM Tris-HCl/150 mM NaCl/10 mM MgCl$_2$/10 mM β-mercaptoethanol/1 mM EGTA/2.5 mM ATP [pH 7]. Aliquots were collected at 0, 2, 24, and 120 h of incubation for immunoblotting. When required, duplicate samples were separated into soluble and pellet fractions using a 20 min centrifugation at 20,000 $g$ before immunoblotting. For NMR analysis, protein samples were incubated in 10 mM Hepes/150 mM NaCl/10 mM MgCl$_2$/1 mM EGTA/2.5 mM ATP [pH 6.9] (phosphorylation buffer).

PLD C415 in LIF buffer was tagged with Alexa Fluor 488 C5 Maleimide (ThermoFisher Scientific) as indicated by the manufacturer. After tagging, 10 equivalents of iodoacetamide (Sigma-Aldrich) were added to prevent further dimer formation of untagged PLD C415. Excess of free dye was removed with Zeba desalting columns. A 70% labeling was confirmed spectrophotometrically.

### Turbidimetry assays

PLD and MetO PLD samples prepared in LIF buffer ranging from 20 to 300 µM were incubated in 20 mM Hepes/150 mM KCl (or 150 mM NaCl when required)/4 mM ATP/4 mM DTT/0.2 mM PMSF/0.03% sodium azide [pH 6.8] (turbidity buffer) supplemented with 30 ng/µl of ethanol-precipitated yeast torula extract RNA (Sigma-Aldrich). Additional measurements using 300 µM proteins (Supplementary Fig. 1D, F) were performed in the presence of 10 mM KCl (NMR buffer) to correlate turbidity and NMR observables. TDP-43 was used at the lower concentration of 10 µM due to its high aggregation propensity. To monitor LLPS in the presence of chaperones and co-chaperones, 20 µM PLD or MetO PLD were incubated with 40 µM of chaperones and 40 µM of co-chaperones in turbidity buffer. Turbidity changes were measured at 25 °C monitoring the absorbance at 600 nm ($A_{600}$) every 15 min for 96 h, with 30 s of agitation (100 rpms) before each measurement in a multiwell plate reader FLUOstar Omega (BMG LABTECH) using 96-well flat-bottom plates (Porvair Sciences). ATP was refreshed every 24 h. All data were technically and biologically replicated ($n$ = at least two biologically independent experiments), and presented as the average between replicates with the standard deviations.

### ThT binding kinetics

Protein samples ranging from 10 to 100 µM were incubated in 10 mM Hepes/150 mM KCl or NaCl/0.03% NaN$_3$ [pH 6.8], containing 10–15 µM ThT (Sigma-Aldrich) (aggregation buffer). Typically, 150 µl samples were placed in wells with a 3 mm glass ball and the kinetics of ThT binding was monitored by bottom reading of fluorescence intensity in a FLUOstar Omega microplate reader (BMG Labtech) at 37 °C as previously described[76]. Measurements were performed using 450 nm excitation and 480 nm emission filters and ten flashes reading every 15 min with 15 s of 200 rpm shaking before reading. All measurements were done in triplicate and the experiments were repeated at least twice using two different protein batches. For seeding experiments, the products of 120 h aggregation reactions were centrifuged 20 min at 20,000 $g$ and the resulting pellets were resuspended in the aggregation buffer by a 15 min sonication at room temperature. The monomer concentration in seed dispersions was determined by SDS-PAGE using fresh PLD and MetO PLD monomer samples as internal controls. Kinetic measurements were carried out using 10 µM of protein monomer solutions containing the sonicated seeds at concentrations corresponding to 0, 0.1, 0.5, and 1% of the monomer concentration. ThT binding to phosphoPLD was performed using 20 µM proteins and a 100:1 ratio of CK1δ in 20 mM Tris-HCl/150 mM NaCl/10 mM MgCl$_2$/10 mM β-ME/1 mM EGTA/2.5 mM ATP [pH 7.0]. Data are presented as average ± SD. When required, the $t_{1/2}$ (time at which half of the monomer is converted into fibril) values were obtained from the fits of sigmoidal curves to the experimental traces, including the SD.

### DIC and fluorescence microscopy

Differential Interference Contrast and fluorescence microscopy images were obtained in a Leica AF6000 LX optical microscope and a Nikon Eclipse TE2000-U microscope, respectively. Protein samples prepared in turbidity buffer (with either 150 mM KCl or 150 mM NaCl) ranging from 20 to 300 µM were incubated at 25 °C for either 3 h (for

PLD and PLD:HSP72:DNAJB1) or 48 h (for PLD, MetO PLD, and PLD:HSP72 and PLD:HSP90β complexes) without agitation. These incubation schemes cover the periods required in most NMR measurements (48 h, see below), demonstrating that condensates remained liquid during the NMR acquisition times. Complex mixtures contained 20 μM PLD, 40 μM HSP70/HSP90, and 40 μM DNAJB1. For fluorescent microscopy, fluorescent-labeled PLD C415 was mixed with untagged PLD at a 1:500 molar ratio before buffer exchange. Samples were spotted onto glass slides (ThermoFisher Scientific) and imaged upright. Imaged samples were replicated.

FRAP experiments were performed on a Leica DMi8 microscope using 150 μM PLD samples mixed 1:1500 with fluorescent-labeled PLD C415 in turbidity buffer. Samples were incubated at 25 °C without agitation for 3, 24, and 96 h. Long incubation times were used to confirm that the condensates present after the typical NMR measurement times (48–96 h) remained fluid. This means that the NMR data report on LLPS and not on solid aggregation. The experiments were performed on static condensates ≥4 μm in diameter, where the bleaching area corresponded to ≤2 μm in diameter. Images were acquired every 0.438 s or 0.657 s with a 63x objective with a numerical aperture of 1.4. Fluorescence intensity was normalized to the unbleached droplet, where the absence of fluorescence intensity corresponded to the area devoid of condensates (background). FRAP bleaching protocols following manufacturer's software reached a minimum of 0.1 normalized fluorescence intensity.

## NMR assignments, $^{15}$N $^{13}$C relaxation analysis, and structure calculation

All the NMR spectra were acquired in 800 MHz ($^1$H) and 600 MHz ($^1$H) Bruker AVNEO spectrometers, both equipped with Z-gradient cryoprobes. $^1$H chemical shifts were referenced to the internal reference sodium trimethylsilylpropanesulfonate (DSS), and $^{15}$N and $^{13}$C chemical shifts were referenced indirectly to $^1$H using the corresponding gyromagnetic ratios[77]. NMR assignments were obtained at 15 °C using 300 μM of $^{15}$N/$^{13}$C-labeled PLD and MetO PLD in NMR buffer, where PLD is significantly de-mixed while MetO PLD remains largely dispersed. These conditions differ substantially from those used previously[23,78]. The following NMR spectra were acquired in the 800 MHz spectrometer to obtain unambiguous assignments: 2D $^{15}$N- and $^{13}$C-HSQCs, 2D CON, 2D CACO, 3D HNCO, HN(CA)CO, HNCA, CBCA(CO)NH, hCC(CO)NH (15 ms mixing time), and HBHA(CO)NH. Additional 3D NMR experiments were acquired on 25, 80, and 150 μM PLD samples in NMR buffer to determine secondary chemical shifts at disperse and de-mixed PLD samples, respectively. MetO PLD$_{309}$ resonance assignments were obtained on a 600 MHz spectrometer at 15 °C on 500 μM samples in NMR buffer, and were based on the acquisition of the following 2D and 3D experiments: 2D $^{15}$N HSQC, 3D HNCO, HN(CA)CO, HNCA, HN(CO)CA, CBCA(CO)NH, HNCACB, and hCC(CO)NH (15 ms mixing time), HNNH, HNHA and $^{15}$N-NOESY-HSQC and $^{13}$C-NOESY-HSQC (both using 120 ms mixing time). Assignments for MetO PLD and MetO PLD$_{309}$ were deposited in the BMRB database (codes 51494 and 34737, respectively).

Loss of NMR signal intensity was calculated by comparing the NMR signal intensities for de-mixed samples (for PLD) at 300 μM vs. the NMR signal intensity for dispersed samples (at 25 μM). Secondary chemical shifts were calculated as ΔCα-ΔCβ, where ΔCα and ΔCβ are the differences between experimentally obtained Cα and Cβ chemical shifts at the specific protein concentrations and computed shifts for disordered PLD at the same temperature and pH (https://spin.niddk.nih.gov/bax/nmrserver/Poulsen_rc_CS/)[79,80]. In the MetO PLD/PLD$_{309}$ plots, secondary chemical shifts for Met residues were removed due to the strong Cβ shifts upon sulfoxidation. Chemical shift perturbation upon methionine sulfoxidation was calculated as follows: $((\delta_{1H}MetO\ PLD - \delta_{1H}PLD)^2 + ((\delta_{15N}MetO\ PLD - \delta_{15N}PLD)/5)_2)^{1/2}$, where the chemical shifts used correspond to the $^{15}$N-HSQC spectra obtained for 300 μM

PLD and MetO PLD, respectively. $^3J_{HNHA}$ coupling constants were calculated on the basis of 3D HNHA experiments[81]. $^{15}$N longitudinal ($R_1$), transverse ($R_2$), and ($^1$H)−$^{15}$N NOE relaxation data for 500 μM MetO PLD$_{309}$ in NMR buffer were obtained using standard Bruker pulse sequences acquired at 15 °C in the 600 MHz spectrometer. In all relaxation experiments, the spectral width was 6579 Hz for $^1$H and 1277 Hz for $^{15}$N dimensions. Seven relaxation delays (20; 60; 150; 240; 460; 800, and 1600 ms) were used for $^{15}$N $R_1$ measurements, while eight relaxation delays (16.96; 33.92; 67.84; 101.76; 152.64; 203.52; 356.16 and 695.36 ms) were used to measure $^{15}$N $R_2$ values. For PLD and MetO PLD, $^{15}$N $R_1$, $R_2$, and ($^1$H)-$^{15}$N NOE relaxation data were acquired on 200 μM samples in NMR buffer at 15 °C in the 800 MHz spectrometer. In all relaxation experiments, the spectral width was 9615 Hz for $^1$H and 1621 Hz for the $^{15}$N dimensions. Seven $^{15}$N $R_1$ relaxation delays were used (100; 200; 300; 400; 600; 800 and 1000 ms) while eight relaxation delays (16.96; 33.92; 67.84; 135.68; 237.43; 339.19; 407.04 and 491.84 ms) were employed for $^{15}$N $R_2$. Relaxation values and uncertainties were calculated by fitting an exponential decay to the data. Het-NOEs were calculated from the ratio of cross peak intensities in spectra collected with and without amide proton saturation during the recycle delay. Uncertainties in peak heights were determined from the standard deviation of the intensity distribution in signal-less spectral regions. $^{15}$N CPMG relaxation dispersion experiments[41] were used to obtain transverse relaxation rates ($^{15}NR_2^{eff}$) as a consequence of the conformational exchange in the μs-ms timescale for 200 μM PLD and MetO PLD in NMR buffer at 15 °C. Each $^{15}$N CPMG relaxation dispersion experiment contained nine interleaved $^{15}$N $B_1$ fields: 0; 33; 133; 200; 267; 533; 733; 866 and 1000 Hz, with a total relaxation delay of 0.06 s in the 800 MHz spectrometer. $^{15}$N CPMG data was analyzed using Nessy[82], where 500 different Monte Carlo sampling simulations were performed using seven independent fits to obtain the exchange rates ($k_{ex}$) and population of the assembled state ($p_B$). Errors were obtained from (1):

$$R_2^{RMSD} = \left\{ \sum \left( R_2^{i,exp} - R_2^{ave} \right)^2 / n \right\}^{1/2} \tag{1}$$

Where $n$ is the number of frequencies, and $R_2^{i,exp}$ and $R_2^{ave}$ are the individual experimental and averaged $R_2$ values, respectively. $k_{ex}$ values for PLD are reported as >2000 s$^{-1}$ to avoid under-interpretation of the CPMG data[83]. The $S^2$ order parameter was derived from the chemical shifts[84]. Sulfoxidized methionines were not included in the $S^2$ plots shown obtained by TALOS-N[85], to avoid over-interpretation of the order parameters due to the absence of sulfoxidized methionines in TALOS libraries. The overall correlation time ($\tau_c$) was estimated from the ratios of the mean values of $T_1$ and $T_2$ as (2):

$$\tau c \simeq 1/(4\pi * {}^{15}N\ frequency\ in\ Hz) * ((6 * R_2/R_1) - 7))^{1/2} \tag{2}$$

which was derived from Eq. (8) of ref. [86] that considers $J(0)$ and $J(\omega)$ spectral densities and discounts terms of higher frequencies from a subset of residues with little internal motion and no significant exchange broadening. This subset excluded residues with $T_2$ values lower than the average minus one standard deviation, unless their corresponding $T_1$ values were larger than the average plus one standard deviation[87]. All NMR spectra were processed in Topspin 4.1.1 and analyzed in Sparky[88].

The NMR structure of MetO PLD$_{309}$ was calculated with the program CYANA v3.98.13[89] based on experimental NOE-derived distance constraints and TalosN-derived dihedral constraints[85] following the standard 7-cycle iterative process and final annealing using the list of restraints obtained in the last cycle. Methionine sulfoxide residues were generated according to CYANA libraries. One hundred structures were generated using the mentioned procedure. Constraint files included 56 upper distance restraints for protons, complemented with

24 $\varphi$ and $\psi$ restrictions. The 20 conformers with the lowest target function values were selected and deposited in the Protein Data Bank under the accession number 8a6i. The structural ensembles were visualized and examined using MolMol[90] and Pymol v2.0 (PyMOL Molecular Graphics System, Version 2.0 Schrödinger, LLC.). PROCHECK-NMR[91] version 3.4.4 was used to analyze the quality of the refined structures. Statistics of the calculation are summarized in Supplementary Table 4.

NMR titrations were based on $^{15}$N-HSQC spectra measured at 15 °C in the 800 MHz spectrometer using 25–35 µM $^{15}$N/$^{13}$C-labeled PLD and MetO PLD in NMR buffer. Increasing amounts of unlabeled (therefore, NMR invisible) chaperones/co-chaperones were added to the titrations. In these conditions, PLD phase separation is negligible. Figures show the data obtained at 1:2 molar ratios for PLD/MetO PLD:chaperones and PLD:co-chaperones, and 1:2:2 for PLD:HSP70:co-chaperones. Due to the abundance of cysteine residues in JDPs, 10 mM DTT was added for their titrations. In addition, 5 mM ZnCl$_2$ was added to the samples containing DNAJA2 to verify the correct fold of the ZnF domain which is present in the protein. A 1:2 ratio was used for the phosphoPLD:HSP70 titrations. The data shown are the reverse of the signal intensity decay for the interactions, which is obtained by comparing the normalized NMR signal intensity for the 1:2 and 1:2:2 complex ratios and the intensity for the PLD alone.

### Circular dichroism (CD)

CD spectra were recorded at 25 °C in a Jasco-810 CD spectrometer with 0.3 mg/ml protein solutions in 5 mM potassium phosphate/10 mM NaCl [pH 6.8] using a 0.1 cm cuvette. For each measurement, six scans were averaged and, after correction for the buffer contribution, transformed into mean residue weight ellipticity ($\Theta_{mrw}$) using 98 as the mean residue molecular weight.

### Transmission electron microscopy (TEM) analysis of negatively stained samples

Samples containing 50 µM PLD and MetO PLD were incubated in turbidity buffer at 37 °C in an orbital shaker operated at 300 rpm. For TEM specimen preparation, 5 µl of 3, 5, 10, and 20 days aged PLD and MetO PLD samples were placed onto carbon-coated, formvar 300 mesh copper grids (Ted Pella) that were freshly glow discharged with a PELCO easiGlow instrument (Ted Pella). After 1 min incubation at room temperature, the solvent excess was blotted with filter paper (Whatman). The grids were next stained with 5 µl of 2% (w/v) uranyl acetate in water for 1 min. After blotting to remove the excess of stain, the grids were dried before imaging. Micrographs were acquired using a Gatan ORIUS SC600 (Model 831) digital camera on a FEI Tecnai G2 20 S-TWIN electron microscope that was operated at 200 kV. Images were replicated four times.

### AFM imaging

Samples containing 100 µM PLD and MetO PLD in turbidity buffer were stored for 4 months at 25 °C with no agitation. 10 µl of 10 µM protein was added on top of a freshly cleaved mica and dried. Imaging was performed using an AFM microscope (Nanotec Electronica) in dynamic mode and HQ:NSC15/No Al cantilevers (µmasch), with a spring constant of 40 N/m and a nominal resonance frequency of 330 kHz. Images were replicated over ten times and were analyzed using WSxM 5.0 software[92].

### Cross-linking assays

Ternary (His-tagged HSP70/DNAJA2/PLD, 40 µM of each protein) and binary (His-tagged HSP70/DNAJA2 and His-tagged HSP70/PLD, 40 µM of each protein) complexes were prepared in 20 mM Hepes/10 mM KCl/5 mM MgCl$_2$/1 mM ATP/1 mM DTT/0.2 mM phenylmethylsulfonyl fluoride (PMSF)/0.03% sodium azide [pH 8.0] and incubated in the absence or presence of 0.025% glutaraldehyde for 10 min. After

quenching with 0.1 M Tris-HCl and 10 mM imidazole (final concentrations), the reaction mixtures were incubated for 2 h under gentle rotatory mixing with fast flow Ni$^{2+}$-loaded Chelating Sepharose$^R$ (Sigma-Aldrich). After several washes with 20 mM Hepes/10 mM KCl/5 mM MgCl$_2$/1 mM ATP/1 mM DTT/0.2 mM PMSF/0.03% sodium azide [pH 7.4], bound His-tagged HSP70 was eluted with 20 mM Hepes/50 mM EDTA/1 mM DTT [pH 8.0] and mixed with Laemmli buffer for immunoblot analysis. Experiments were repeated twice.

### SDS-PAGE and immunoblot

Samples containing 10 µM proteins prepared in Laemmli buffer with β-mercaptoethanol (BioRad) and heated at 95 °C for 15 min were separated by SDS-PAGE using Mini-Protean® TGX Stain-Free™ gels (#456-8126 or #456-1026, BioRad). If needed, gels were stained with Coommassie Brilliant Blue R-250 staining solution (BioRad) for their imaging and quantitative analysis using a ChemiDoc XRS-Plus imager (BioRad). When required, electrotransference onto 0.2 µm immunoblot PVDF membranes was performed in Towbin buffer (25 mM Tris/192 mM Gly [pH 8.3] with 10% methanol) for 50–90 min at 4 °C. After blocking for 1 h in TBS (20 mM Tris-HCl/150 mM NaCl [pH 7.5]) containing 0.5% (v/v) Tween 20 (TBS-T) and 0.5% (w/v) bovine serum albumin (Sigma-Aldrich) at room temperature, the membranes were incubated overnight at 4 °C with agitation with the mouse monoclonal anti-TDP-43 3H8 (Novus, diluted 1:5000 in blocking buffer), the rabbit monoclonal anti-phospho Ser410 TDP-43 (Sigma-Aldrich, diluted 1:1000 in blocking buffer), the rabbit polyclonal anti-DNAJA2 (Proteintech, diluted 1:1000 in blocking buffer) or with HisProbe-HRP (ThermoFisher Scientific, diluted 1:2000 in blocking buffer) antibodies. The membranes were washed three times for 10 min each with TBS-T and then incubated with either horseradish peroxidase (HRP)-labeled goat anti-mouse IgG (Sigma-Aldrich, 1:5000 dilution) for 3H8, or anti-rabbit IgG (Sigma-Aldrich, 1:4000 dilution) for anti-phospho Ser410 and anti-DNAJA2. After washing, the membranes were developed using Clarity Western-ECL (BioRad), and the signals were recorded with the ChemiDoc XRS-Plus imager (BioRad). Immunoblots were repeated twice. Uncropped gels were included in the Source Data File.

### Protein solubility determination

PLD and MetO PLD samples ranging 20–200 µM were incubated at 15 °C in NMR buffer (NMR conditions) and at 25 °C in turbidity buffer with agitation (turbidity conditions). After a clarification step of 20,000 $g$ for 30 min, 2 µg of the samples were analyzed by SDS-PAGE. At least two replicates of each sample were analyzed. The quantification process was done by the Stain-Free imaging Bio-Rad technology using a Bio-Rad ChemiDoc™ Imaging System and SDS-PAGE gels containing trihalo compounds.

### Reporting summary

Further information on research design is available in the Nature Portfolio Reporting Summary linked to this article.

## Data availability

All materials are readily available from the corresponding author upon request. The structural data generated in this study have been deposited in the PDB database under the accession code 8a6i and the chemical shifts data in the BMRB database under the accession codes 51494 and 34737. Source data are provided with this paper.

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

## Acknowledgements

This work was supported by grants PID2019-109276RA-I00/AEI/ 10.13039/501100011033 (to J.O.), PID2019-103845RB-C21-AEI/10.13039/ 501100011033 (to M.G.) and SAF2016-76678-C2-2-R (MINECO/AEI/ FEDER UE to D.V.L.) from the Spanish AEI-Ministry of Science and Innovation and Enhanced New Staff Start-up Research Grant (The University of Hong Kong) to R.H. J.O. was a recipient of a Leonardo Grant from the Spanish BBVA Foundation (BBM_TRA_0203) and is a Ramón y Cajal Fellow of the Spanish AEI-Ministry of Science and Innovation (RYC2018-026042-I funded by MCIN/AEI/10.13039/501100011033 and by "ESF Investing in your future"). NMR experiments were performed in the "Manuel Rico" NMR Laboratory (LMR) of the Spanish National Research Council (CSIC), a node of the Spanish Large-Scale National Facility (ICTS R-LRB). Authors would like to acknowledge Prof. Emanuele Buratti (International Centre for Genetic Engineering and Biotechnology, Trieste, Italy) for kindly providing the TDP-43 cDNA clone, Dr. José Manuel Pérez Cañadillas (IQFR/CSIC) for providing the TXA fusion plasmid, Dr. María José Sánchez-Barrena (IQFR/CSIC) for providing the His-Probe HRP, the use of Servicio de Microscopía Láser Confocal y Multidimensional in vivo (CIB-CSIC), the Proteomics Facility (CNB-CSIC), to Dr. Pilar Lillo and Dr. Carolina García for assistance with the fluorescence microscopy (IQFR-CSIC) and to M.G. Mateu (CBMSO-CSIC/UAM) for granting AFM time and materials for this study. We are grateful to Dr. Alberto García Redondo (Hospital Doce de Octubre) for critical reading of the manuscript.

## Author contributions

Conceptualization, M.G., J.O. Sample preparation, data acquisition and analysis, J.C., R.A., M.G., J.O. NMR structure calculation, D.P.U., J.O. AFM imaging, A.V. TEM image acquisition, M.M., R.H. Writing—Original Draft, J.O. Writing, Review and Editing, all authors. Funding acquisition, D.V.L., M.G., J.O. Supervision, M.G., J.O.

## Competing interests

J.C., R.A., D.P.U., D.V.L., M.G., and J.O. are co-inventors of the patent application *EP22383245 filed on 20/12/2022 by Consejo Superior de Investigaciones Científicas, that includes the production, sulfoxidation and conformational features of PLD and PLD$_{309}$ fragments.* A.V., M.M., and R.H. declare no competing interests.
