## [Peer Review File · Nature Communications]

REVIEWER COMMENTS

Reviewer #1 (Remarks to the Author):

The authors have characterized the effects of chaperones, co-chaperones and oxidation on TDP-43 aggregation and phase separation. Using complementary biophysics methods (Fluorescence, DIC, ThT and turbidity assays) and structural biology approaches (NMR, AFM, TEM), authors have shown that methionine oxidation reduces helical content of TDP-43, inhibits chaperones binding and phosphorylation, and affects phase separation and amyloid formation. Understanding the effects of protein post-translational modifications on chaperon interaction and amyloid formation is particularly important and challenging. Nevertheless some of the conclusions, drawn from NMR data, are not supported by convincing data and a major revision, including new data and quantitative analysis, is required before that the manuscript become publishable in Nature communication.

Major Point: NMR signals broadening is observed in the region 305-345 upon phase separation. Such effect can be due to a decrease of local flexibility (i.e. increase of order parameter S^2), or self-association (increase of overall tumbling time τ_C), or presence of conformational/chemical exchange in the micro- to millisecond time scale. The authors concluded that this broadening is due to intermolecular interaction and compare this signal broadening for different forms of PLD (unmodified, oxidation) or addition of chaperones. Such conclusions are not supported by adequate NMR data and analysis. The variation of the broadening can be due to various key factors such as a change in exchange process (variation of population or exchange constant), change in S^2 or apparent τ_C , each of these factors corresponding a different molecular event. A rigorous quantitative investigation of order parameters, apparent τ_C and exchange parameters (k_{ex} and P) using already acquired relaxation experiments and supplemented by Relaxation Dispersion experiments is a prerequisite before to draw conclusion from observed line broadening.

Minor Points:

1. Top of page 9: Longer and unbranched fibrils are observed with MetO PLD suggesting alternative packing, according to authors. Such change can be due to an inhibition of secondary nucleation mechanism. The authors should address this point.
2. Fig.3: For clarity reason, comparison with data acquired with unmodified PLD should be presented, at least in supporting information.
3. Page 5, lines 210-213: "lower values for segment 326-332" does not match with fig. 3C, where lower values are observed for residues 322 to 328. Similarly the green cylinder representing the transient helix is not aligned with low value of J-couplings (3.C) and observed secondary chemical shifts (3.B)
4. Fig3: error bars should be provided for panels A and C.
5. The data reported in figure 3 clearly indicate a helical propensity populated at ca. 27%. In absence of structurally meaningful inter-residue NOE distance restraints, more caution should be taken in the analysis of structural models. With such a low number of structural restraints the structure statistic are not very relevant.
6. Fig4G: The similarity of PLD NMR signals intensities in presence of different DNAJ and HSP70 do not seem to be compatible with the formation of a ternary complex. Lost in intensity is expected upon binding of HSP70. The authors should address this point.
7. Supporting Figure 9 A and 9B: The authors should present the full rectangular images.

Reviewer #2 (Remarks to the Author):

This manuscript describes the effect of methionine sulfoxidation on the liquid-liquid phase separation (LLPS) behavior and aggregation properties of the low-complexity (prion-like) domain of TDP-43. The authors propose that methionine sulfoxidation greatly impairs the capacity of the protein to undergo LLPS as well as aggregation into amyloid fibrils. It also affects the interaction of TDP-43 with a range of molecular chaperones. The general topic of this study is certainly of interest. However, there are a number of problems with experimental data and their interpretation. This, together with the lack of appropriate controls make the conclusions less than compelling.

1. The first major claim of the manuscript is that methionine sulfoxidation greatly impairs protein propensity for LLPS. While this is possible, this central claim needs much better documentation. Data presented by the authors in this regard are largely limited to turbidimetry measurements under very limited set of conditions. Given that turbidity increase can result both from LLPS (droplet formation) as well as protein aggregation, this claim needs to be supported by technically convincing light microscopy data (preferentially fluorescence microscopy) and phase diagrams (or other types of data presentation) to determine saturation concentrations of unmodified and modified proteins under the experimental conditions used by the authors. Turbidimetry and light microscopy data shown in Fig. 1 B and C are unconvincing and likely misinterpreted, as the heterogeneous structures shown in upper panel of Fig. 1C look more like protein aggregates rather than classical liquid droplets. It is critical that LLPS and liquid-like character of the droplets are confirmed by FRAP measurements which are a standard approach in LLPS research. Also, no microscopy data are shown for the modified protein to validate the claim that this protein does not undergo LLPS.

2. Information regarding saturation concentration is also crucial to support the claim regarding the same helical propensity of the protein in the dilute and condensed phases.

3. The CD spectrum shown in Fig. S6 for unmodified protein suggests rather high helical content and appears to be significantly different from those shown in other studies [e.g., Lim et al., PLoS Biol 14, e1002338 (2016)]. Could the authors estimate the helical content from their CD spectrum and compare it with that deduced from NMR data?

4. Given that high local protein concentration within droplets greatly increases aggregation kinetics, in order to make any meaningful claim regarding the effect of methionine sulfoxidation on fibrillation propensity of TDP-43, experiments need to be done under conditions where both modified and non-modified proteins are in the dilute phase (i.e., no LLPS). This is not the case for experiments shown in Fig. 6 at a protein concentration of 100 μ M, where large proportion of unmodified (but not modified) protein is likely in the condensed phase.

5. The claim that oxidized protein assembles into structurally different fibrils is based solely on a rather crude morphological data which provide no information about packing arrangements within fibril core. Thus, at the very least, the conclusions based on these morphological observations should be toned down.

6. With regard to the experiments on the impact of molecular chaperones on TDP43 LLPS, these data appear somewhat preliminary. Again, more conclusive evidence should be provided that turbidity changes reflect solely LLPS and not protein aggregation. Furthermore, the observations on distinct effect of different chaperones are very descriptive in nature, with no mechanistic insight into these apparent differences.

Reviewer #3 (Remarks to the Author):

In the paper "Metamorphism in TDP-43 prion-like domain determines chaperone recognition" the authors characterized the effect of Methionine oxidation on the conformational, phase separation, fibrillation, chaperone recognition and phosphorylation properties of the prion-like domain of TDP-43 (PLD). This low complexity region comprises the stretch of residues 274-414 which is rich in Met residues and promotes the aggregation of TDP-43.

Salient features of this work include:

- 1) Methionine oxidation prevent de-mixing of PLD by LLPS**
- 2) It does so by preventing self-association of an hydrophobic stretch comprising residues 320-342 and by decreasing helicity at this region**
- 3) Oxidized PLD impairs canonical fibril formation, introducing alternative aggregation pathways with morphologically distinct aggregates**
- 4) Cytosolic chaperones, such as HSP90 and HSP70 and others, promote de-mixing of PLD by direct interactions with the hydrophobic region**
- 5) Methionine oxidation suppresses chaperone recognition of the PLD**
- 6) Methionine oxidation prevents phosphorylation of PLD serine residues by CK1d**

These in vitro results were integrated with current knowledge about the cellular behavior of TDP-43 to propose different protein outcomes upon cellular stress as observed during the onset of ALS and other dementias. The work is very interesting and provides novel insights about the role of methionine oxidation on protein conformations, phase separation and aggregation propensities. I think it would be of great interest for the general audience of Nat. Commun.

There are a few minor points that I feel should be clarified and complemented:

a) The de-mixing process takes quite some amount of time. Incubations span days. The 1H-15N HSQC spectrum of de-mixed PLD (Figure 1E and Supplementary Figures 1 and 2) show a large number of minor (unassigned) cross-peaks and some of them (between 7.6-8 ppm in 1H and around 125 ppm in 15N) could arise from new C-termini due to limited proteolysis (even in the presence of pmsf in the buffer, which has a limited lifetime in aqueous solutions). If that's the case, there might be different species separating differently. It would be helpful if the authors show an SDS-PAGE and/or a western blot of PLD and MetO PLD after de-mixing.

b) For figure 3 it would be nice to include NMR data on reduced PLD309 under experimental conditions similar to those used for MetO PLD309 for direct comparisons. I realize these were previously presented by another group in Structure, 24, 1537, but including it will better highlight the effects of Met oxidation on the dynamics of PLD309.

c) The phosphorylation results are not as straightforward as the rest of the manuscript. Mainly because the authors do not present quantitative determinations of the levels of phosphorylation of the different PLDs used. Accordingly, it is difficult to say how much of the effects are due to phosphorylation or to the remnants of unmodified PLDs. Can the authors determine the levels of phosphorylation by NMR or mass spectrometry? Phosphoserine cross-peaks of 15N labeled proteins (such as those used here) have easily recognized, distinct chemical shifts that may shed some light on this. Or maybe phosphorus NMR.

d) In the same line, the authors showed that phosphorylation of PLD was "drastically

diminished” when PLD Met residues are oxidized and the reason for this would be that oxidation interferes with the client recognition of CK1d. This is certainly one explanation but another one, with a similar rationale, would be that oxidation of nearby Met side chains (for instance M405 and M414) might alter the epitope recognition of pS410 antibody. The antibody was raised against a reduced pSer410 substrate so Met side chain oxidation could lead to less binding to the blotted proteins and hence less amount of signal. Detection of PLD phosphorylation by NMR may also help here. If precipitation induced by phosphorylation is a problem, dissolving modified aggregates in urea and comparing with denatured unmodified references could also help.

REVIEWER COMMENTS

Reviewer #1 (Remarks to the Author):

The authors have characterized the effects of chaperones, co-chaperones and oxidation on TDP-43 aggregation and phase separation. Using complementary biophysics methods (Fluorescence, DIC, ThT and turbidity assays) and structural biology approaches (NMR, AFM, TEM), authors have shown that methionine oxidation reduces helical content of TDP-43, inhibits chaperones binding and phosphorylation, and affects phase separation and amyloid formation. Understanding the effects of protein post-translational modifications on chaperon interaction and amyloid formation is particularly important and challenging. Nevertheless some of the conclusions, drawn from NMR data, are not supported by convincing data and a major revision, including new data and quantitative analysis, is required before that the manuscript become publishable in Nature communication.

We are very grateful to Reviewer 1 for the overall favorable evaluation and for the constructive criticism. We have taken these concerns to heart and performed several additional experiments, which have led to four new figures in the revised manuscript. These new findings now provide, in our opinion, convincing data to support the conclusions advanced in the original version of the manuscript. They are described in detail below.

Major Point: NMR signals broadening is observed in the region 305-345 upon phase separation. Such effect can be due to a decrease of local flexibility (i.e. increase of order parameter S^2), or self-association (increase of overall tumbling time τ_C), or presence of conformational/chemical exchange in the micro- to millisecond time scale. The authors concluded that this broadening is due to intermolecular interaction and compare this signal broadening for different forms of PLD (unmodified, oxidation) or addition of chaperones. Such conclusions are not supported by adequate NMR data and analysis. The variation of the broadening can be due to various key factors such as a change in exchange process (variation of population or exchange constant), change in S^2 or apparent τ_C , each of these factors corresponding a different molecular event. A rigorous quantitative investigation of order parameters, apparent τ_C and exchange parameters (k_{ex} and P) using already acquired relaxation experiments and supplemented by Relaxation Dispersion experiments is a prerequisite before to draw conclusion from observed line broadening.

We are thankful to the reviewer for these comments and agree that obtaining the mentioned relaxation parameters would be very valuable towards the interpretation of the data. We have acquired additional relaxation experiments for the PLD and MetO PLD in LLPS conditions and obtained the required parameters. As shown in **Supplementary Fig. 6** and **Supplementary Table 2** in the revised manuscript, ^{15}N spin relaxation rates provide clear evidence of intermolecular associations for the PLD in LLPS conditions. In particular, we have calculated the apparent τ_C from the ratio of the mean values of T_1 and T_2 (specifically, for the region where we observe line broadening and secondary structure propensities, covering residues 321-343) and show that the overall correlation time for the PLD doubles that for MetO, which serves as a strong indication that the PLD is assembling into larger species. Moreover, the overall higher values for $(^1\text{H})\text{-}^{15}\text{N}$ heteronuclearNOEs for the PLD compared to MetO PLD indicate that the PLD is more rigid. The S^2 order parameters for both proteins cannot be confidently calculated applying Lipari-Szabo ModelFree parameters due to the short and low-populated structured elements in both proteins, and to the low intensity of the heteronuclear NOES (<0.5 for the PLD). Nevertheless, we have calculated the S^2 order parameters from the chemical shifts as described by (Berjanskii and Wishart (2005) JACS **127**: 14970-1) and show that the values are significantly higher for the PLD, in agreement with the loss of flexibility determined by $(^1\text{H})\text{-}^{15}\text{N}$ heteronuclearNOEs (**Supplementary Fig. 6**). For the signal broadening analysis shown in **Fig. 1F**, we compared the NMR signal intensity for the proteins in high vs. low concentrations. As shown in **Supplementary Fig. 7**, the structured elements remain identically populated in both conditions for the PLD. Therefore, the NMR signal line broadening does not emerge from a decrease in local flexibility at higher concentrations due to changes in the conformation, but rather from self-association (as indicated by the increased correlation time).

More importantly, our new ^{15}N CPMG relaxation dispersion data shows fast conformational exchange between species ($k_{ex} = >2000 \text{ s}^{-1}$) for the PLD, while MetO PLD shows

lack of exchange. Overall, the relaxation data confirms that the PLD assembles into larger species *via* the double α -helix in a very dynamic and concentration-dependent fashion as indicated by NMR signal broadening (**Fig. 1F**), while MetO PLD remains highly flexible and largely disperse in solution even at high concentrations.

Our data largely agree with the conclusions drawn in Conicella *et al.* (2016) *Structure*, **24** 1537-1549; for a very similar protein construct although under different experimental conditions. In that study, the authors concluded that TDP-43's PLD assembles into dynamic condensates *via* the α -helical region. In addition, it is well established that disordered proteins show significant NMR signal broadening upon intermolecular associations (for comparison, see Oroz *et al.* (2018) *Nature Communications*, **9**: 4532). Based on these new results and considerations, we are confident that the NMR signal broadening observed for the PLD in LLPS conditions arises from intermolecular associations involved in LLPS.

Minor Points:

1. Top of page 9: Longer and unbranched fibrils are observed with MetO PLD suggesting alternative packing, according to authors. Such change can be due to an inhibition of secondary nucleation mechanism. The authors should address this point.

We agree with the reviewer that determining the impact of methionine sulfoxidation in the nucleation mechanisms is highly interesting. We have acquired new kinetic aggregation data (shown in new **Supplementary Figure 9**). In brief, the data show that the aggregation of the PLD and MetO PLD is governed by distinct mechanisms, due to their different dependence of the aggregation kinetics with the monomer concentration. In particular, the lack of a lag phase and of a concentration effect in the $t_{1/2}$ suggests that the limiting step in the aggregation of the PLD are the conformational transitions towards amyloid-compatible structures (see Leonil *et al.* (2008) *J Mol Biol* **381**: 1267-80). On the contrary, methionine sulfoxidation severely impairs primary nucleation, but not secondary nucleation (see Thacker *et al.* (2020) *PNAS* **117**: 25272-83), as can be concluded from the significant reduction of the $t_{1/2}$ upon increasing concentrations of fibril seeds (see new **Supplementary Table 3**). Therefore, we believe that the statement regarding the alternative packing in MetO PLD fibrils is valid. This new evidence is included in the revised version of the manuscript.

2. Fig.3: For clarity reason, comparison with data acquired with unmodified PLD should be presented, at least in supporting information.

We believe that the comparison of the ^{15}N spin relaxation parameters of the longer PLD and MetO PLD included in the new **Supplementary Fig. 6** conveniently addresses this issue.

3. Page 5, lines 210-213: "lower values for segment 326-332" does not match with fig. 3C, where lower values are observed for residues 322 to 328. Similarly the green cylinder representing the transient helix is not aligned with low value of J-couplings (3.C) and observed secondary chemical shifts (3.B).

We appreciate this comment from the reviewer and have corrected the inconsistencies. The green cylinder on top of the plots is aligned with the α -helix obtained by CYANA for MetO PLD₃₀₉ (covering residues 324-332).

4. Fig3: error bars should be provided for panels A and C.

For clarity, we have modified the colors of the error bars (now in black). In most cases the bars are so small that are included inside the marker spheres.

5. The data reported in figure 3 clearly indicate a helical propensity populated at ca. 27%. In absence of structurally meaningful inter-residue NOE distance restraints, more caution should be taken in the analysis of structural models. With such a low number of structural restraints the structure statistic are not very relevant.

We agree with the reviewer that the structure of MetO PLD₃₀₉ is hampered by the lack of meaningful NOEs, due to its high flexibility. Interpretation of NOE data for small, dynamic peptides

can be complicated due to fast conformational averaging, since peptides are expected to sample a number of backbone conformations on a nanosecond time scale. The NOESY spectrum will contain crosspeaks representative of all conformations that are sufficiently populated. However, NMR parameters such as chemical shifts and coupling constants are population-weighted averaged over all conformers, and thus represent a valid data source for low-populated structured peptides (Wright PE *et al.* (1988) **27**: 7167-75). In addition, the structure has been validated by PROCHECKNMR, showing that all the angles encompassed in the structure are present in favored regions of the Ramachandran plot (**Supplementary Table 4**, certified in the PDB Structure Validation Report).

6. Fig4G: The similarity of PLD NMR signals intensities in presence of different DNAJ and HSP70 do not seem to be compatible with the formation of a ternary complex. Lost in intensity is expected upon binding of HSP70. The authors should address this point.

The reviewer raises an interesting point. Because HSP70 and DNAJs mainly interact with the double α -helical region of the PLD, addition of HSP70 to a pre-formed PLD:DNAJ complex would show differences in this region if HSP70 would remove the PLD from DNAJ. Because we do not see differences in this region of the PLD in the NMR intensity plots, we conclude that the PLD remains bound to DNAJ using the α -helical region, even in the presence of HSP70. However, we see additional regions from the PLD involved in complex formation once HSP70 is added to the PLD:DNAJ complex, especially evident in the region 310-320 (where the LARKS is located, which may be relevant for chaperone recognition due to the presence of two nearby Phe residues: F313 and F316) and the region 370-380 and around 400 (probably due to the presence of F367, Y374, W385, F397 and F401). For more on the role of aromatics in chaperone client recognition see: Karagöz, *et al.* (2014) *Cell* **156**: 963-74. Therefore, we conclude that NMR provides evidences of ternary complex formation. This discussion is included in the revised manuscript.

7. Supporting Figure 9A and 9B: The authors should present the full rectangular images.

We thank the reviewer for raising this point. The new **Supplementary Figure 12** (corresponding to **Supplementary Figure 9** in the original version of the manuscript) includes the full rectangular image.

Reviewer #2 (Remarks to the Author):

This manuscript describes the effect of methionine sulfoxidation on the liquid-liquid phase separation (LLPS) behavior and aggregation properties of the low-complexity (prion-like) domain of TDP-43. The authors propose that methionine sulfoxidation greatly impairs the capacity of the protein to undergo LLPS as well as aggregation into amyloid fibrils. It also affects the interaction of TDP-43 with a range of molecular chaperones. The general topic of this study is certainly of interest. However, there are a number of problems with experimental data and their interpretation. This, together with the lack of appropriate controls make the conclusions less than compelling.

1. The first major claim of the manuscript is that methionine sulfoxidation greatly impairs protein propensity for LLPS. While this is possible, this central claim needs much better documentation. Data presented by the authors in this regard are largely limited to turbidimetry measurements under very limited set of conditions. Given that turbidity increase can result both from LLPS (droplet formation) as well as protein aggregation, this claim needs to be supported by technically convincing light microscopy data (preferentially fluorescence microscopy) and phase diagrams (or other types of data presentation) to determine saturation concentrations of unmodified and modified proteins under the experimental conditions used by the authors. Turbidimetry and light microscopy data shown in Fig. 1 B and C are unconvincing and likely misinterpreted, as the heterogeneous structures shown in upper panel of Fig. 1C look more like protein aggregates rather than classical liquid droplets. It is critical that LLPS and liquid-like character of the droplets are confirmed by FRAP measurements which are a standard approach in LLPS research. Also, no microscopy data are shown for the modified protein to validate the claim that this protein does not undergo LLPS.

We thank the reviewer for her/his constructive critics of the manuscript. In the new version of the manuscript, we have included additional FRAP measurements, phase diagrams and protein quantification in the different phases. In particular, the FRAP experiments included in **Supplementary Figure 1D** show that the condensates formed by the PLD are highly fluid even after 96h of incubation (longer incubation time than the NMR relaxation experiments presented in **Supplementary Figure 6**). This remarkable fluidity within the condensates correlates with the high exchange determined by ^{15}N CPMG relaxation dispersion for the PLD in LLPS conditions (**Supplementary Figure 6D**). This evidence, in combination with the high resolution DIC images presented in the manuscript (**Figs. 1C, 4B, 4F**), allows us to conclude that turbidity arises from LLPS and not protein aggregation. In this regard, the phase diagrams shown in **Supplementary Figure 1A** clearly demonstrate that methionine sulfoxidation strongly impairs LLPS. **Supplementary Figure 15B** shows that LLPS is impeded by methionine sulfoxidation even at high protein concentrations, as monitored by DIC microscopy. Furthermore, high-resolution protein quantification shows that only ~35% of the PLD remains in the soluble fraction after the NMR experiments, while no MetO PLD is detected in the de-mixed fraction, after clarification at 20,000 g for 30 minutes (**Supplementary Fig. 1C**). All in all, the data demonstrate that the PLD undergoes significant LLPS in the NMR conditions forming fluid condensates, which is strongly diminished by methionine sulfoxidation.

2. Information regarding saturation concentration is also crucial to support the claim regarding the same helical propensity of the protein in the dilute and condensed phases.

Supplementary Figure 7 shows that the population of α -helical conformers for the PLD at 25 μM is comparable to 300 μM . Protein quantification shows that at the lower range of protein concentrations, almost all PLD protein remains in the disperse phase (~95%, **Supplementary Fig. 1C**). On the other hand, the PLD at the higher range of protein concentrations undergoes significant LLPS, and only ~35% of the PLD remains soluble at 200 μM . However, MetO PLD remains largely disperse even at high protein concentrations. Therefore, we conclude that the statements included in **Fig. 1G** regarding disperse and de-mixed phases are valid.

3. The CD spectrum shown in Fig. S6 for unmodified protein suggests rather high helical content and appears to be significantly different from those shown in other studies [e.g., Lim et al., *PLoS Biol* 14, e1002338 (2016)]. Could the authors estimate the helical content from their CD spectrum and compare it with that deduced from NMR data?

We thank the reviewer for pointing out this difference. It must be noted that the polypeptide chains and buffers used in both studies are different. Lim et al. (2016) used the 263-414 fragment with a six His tag at the C-terminus prepared in 1 mM phosphate buffer [pH 6.8]. Moreover, 20 μM of protein were used and the helical content increased over incubation times. Here, we have used 0.3 mg/ml (aprox. 20 μM) of the His-tag cleaved 274-414 fragment freshly prepared in 5 mM potassium phosphate/10 mM NaCl [pH 6.8] and measured at 25 $^{\circ}\text{C}$. Helix content can be estimated from the mean residue ellipticity at 222 nm ($[\theta]_{222}$) using the equation $\% \alpha\text{-helix} = 100 ([\theta]_{222} / (-39500(1 - 2.57/n)))$, where n is the number of total peptide bonds (140 in our case) (Sommese *et al.* (2010) *Prot Sci* **19**: 2001-5). This yields an α -helix content of 12%, while the NMR data (obtained averaging the $\text{C}\alpha$ chemical shifts, considering 3.1 ppm $\text{C}\alpha$ chemical shifts for 100 % α -helical conformers) yield around 40% for the α -helix spanning residues 321-331 and 10 % for the α -helix spanning residues 334-343. These values are in agreement with the ones reported by Conicella *et al.* (2016) *Structure* **24**, 1537-49. Overall, the NMR measurements obtained at 15 $^{\circ}\text{C}$ for the PLD report an averaged 25,15% of α -helical content, which is higher than the content reported by CD at 25 $^{\circ}\text{C}$. This difference may be attributed in part to the higher temperature of the CD measurements. In addition, it is well known (Chakrabarty *et al.* (1993) *Biochemistry* **32**: 5560-5565), that aromatic residues contribute bands to the far UV CD spectra which make the estimation of α -helical population less reliable. Since the effect of aromatic residues on $^{13}\text{C}\alpha$ chemical shifts is much weaker, we consider that the values measured by NMR are more accurate and precise. These differences are mentioned in the modified version of the manuscript.

4. Given that high local protein concentration within droplets greatly increases aggregation

kinetics, in order to make any meaningful claim regarding the effect of methionine sulfoxidation on fibrillation propensity of TDP-43, experiments need to be done under conditions where both modified and non-modified proteins are in the dilute phase (i.e., no LLPS). This is not the case for experiments shown in Fig. 6 at a protein concentration of 100 μ M, where large proportion of unmodified (but not modified) protein is likely in the condensed phase.

Fig. 5C includes the aggregation kinetics obtained for 20 μ M PLD and MetO PLD. In addition, **Supplementary Fig. 9** includes the aggregation data for the PLD and MetO PLD at 9-20 μ M protein concentration. Both proteins remain in the disperse phase at these protein concentrations (**Supplementary Fig. 1C**). Even at this concentration range, where both proteins remain soluble, the aggregation kinetics for MetO PLD is significantly delayed, likely because the primary nucleation is strongly affected (**Supplementary Fig. 9**).

5. The claim that oxidized protein assembles into structurally different fibrils is based solely on a rather crude morphological data which provide no information about packing arrangements within fibril core. Thus, at the very least, the conclusions based on these morphological observations should be toned down.

We agree with the reviewer and have qualified the claims regarding the alternative packing of the MetO PLD within the fibrils to avoid any overinterpretation of the fibrils morphological description.

6. With regard to the experiments on the impact of molecular chaperones on TDP43 LLPS, these data appear somewhat preliminary. Again, more conclusive evidence should be provided that turbidity changes reflect solely LLPS and not protein aggregation. Furthermore, the observations on distinct effect of different chaperones are very descriptive in nature, with no mechanistic insight into these apparent differences.

We agree with the reviewer on the interest of understanding the impact of chaperones on TDP-43 LLPS. Indeed, a discussion regarding the molecular basis of the differential effects observed for the different HSP40 isoforms on TDP-43 LLPS is included in the manuscript. Nonetheless, the scope of this study is to understand the structural changes triggered by methionine sulfoxidation on TDP-43 PLD, and how these changes in structure and dynamics alter LLPS, chaperone recognition, amyloid aggregation and phosphorylation. There is a remarkable interest in deciphering the differential roles of the large HSP40 family in the clearance of aggregating proteins, which may be relevant in several neurodegenerative diseases (see for instance Mok *et al.* (2018) *Nat Struct Molec Biol* **25**: 384-393; Hou *et al.* (2021) *Nat Commun* **12**: 5338; Rozales *et al.* (2022) *Nat Commun* **13**: 516). We believe that the conclusions derived from our observations will raise awareness on the relevance of post-translational modifications in the triage of client proteins by chaperones in future studies.

Reviewer #3 (Remarks to the Author):

In the paper "Metamorphism in TDP-43 prion-like domain determines chaperone recognition" the authors characterized the effect of Methionine oxidation on the conformational, phase separation, fibrillation, chaperone recognition and phosphorylation properties of the prion-like domain of TDP-43 (PLD). This low complexity region comprises the stretch of residues 274-414 which is rich in Met residues and promotes the aggregation of TDP-43.

Salient features of this work include:

- 1) Methionine oxidation prevent de-mixing of PLD by LLPS
- 2) It does so by preventing self-association of an hydrophobic stretch comprising residues 320-342 and by decreasing helicity at this region
- 3) Oxidized PLD impairs canonical fibril formation, introducing alternative aggregation pathways with morphologically distinct aggregates

4) Cytosolic chaperones, such as HSP90 and HSP70 and others, promote de-mixing of PLD by direct interactions with the hydrophobic region

5) Methionine oxidation suppresses chaperone recognition of the PLD

6) Methionine oxidation prevents phosphorylation of PLD serine residues by CK1d

These in vitro results were integrated with current knowledge about the cellular behavior of TDP-43 to propose different protein outcomes upon cellular stress as observed during the onset of ALS and other dementias. The work is very interesting and provides novel insights about the role of methionine oxidation on protein conformations, phase separation and aggregation propensities. I think it would be of great interest for the general audience of Nat. Commun.

We are thankful to the reviewer for her/his very positive comments.

There are a few minor points that I feel should be clarified and complemented:

a) The de-mixing process takes quite some amount of time. Incubations span days. The 1H-15N HSQC spectrum of de-mixed PLD (Figure 1E and Supplementary Figures 1 and 2) show a large number of minor (unassigned) cross-peaks and some of them (between 7.6-8 ppm in 1H and around 125 ppm in 15N) could arise from new C-termini due to limited proteolysis (even in the presence of pmsf in the buffer, which has a limited lifetime in aqueous solutions). If that's the case, there might be different species separating differently. It would be helpful if the authors show an SDS-PAGE and/or a western blot of PLD and MetO PLD after de-mixing.

New **Supplementary Figure 1C** shows an SDS-PAGE containing 200 μ M PLD and MetO PLD samples after the NMR experiments. New **Supplementary Figs. 17, 18** show western blots containing 20 μ M PLD and MetO PLD after the incubation for 24-120 h at 37 °C. No significant degradation products were visible.

b) For figure 3 it would be nice to include NMR data on reduced PLD309 under experimental conditions similar to those used for MetO PLD309 for direct comparisons. I realize these were previously presented by another group in Structure, 24, 1537, but including it will better highlight the effects of Met oxidation on the dynamics of PLD309.

We agree with the reviewer in the interest of comparing the relaxation parameters of PLD and MetO PLD. New **Supplementary Figure 6** shows the ¹⁵N spin relaxation parameters for PLD and MetO PLD. We believe that the differences observed in this data are significant enough for a valid comparison and highlight the effects of methionine sulfoxidation on the dynamics of the PLD: MetO PLD remains highly flexible even at high concentrations, while PLD assembles into larger species via the double α -helical motif.

c) The phosphorylation results are not as straightforward as the rest of the manuscript. Mainly because the authors do not present quantitative determinations of the levels of phosphorylation of the different PLDs used. Accordingly, it is difficult to say how much of the effects are due to phosphorylation or to the remnants of unmodified PLDs. Can the authors determine the levels of phosphorylation by NMR or mass spectrometry? Phospho-serine cross-peaks of ¹⁵N labeled proteins (such as those used here) have easily recognized, distinct chemical shifts that may shed some light on this. Or maybe phosphorus NMR.

NMR has proven a very powerful tool to determine phosphorylation rates in IDPs (see Schwalbe *et al.* (2013) Biochemistry **52**: 9068-9079). Unfortunately, CK1 δ phosphorylation promotes the fast aggregation of the PLD, and no phosphoserine moieties are detectable in the NMR spectra. This fast aggregation will impede any quantitative determination by solution state NMR. Yet, we observed that methionine sulfoxidation impaired CK1 δ phosphorylation, which is evident from the NMR and blotting data.

d) In the same line, the authors showed that phosphorylation of PLD was "drastically diminished"

when PLD Met residues are oxidized and the reason for this would be that oxidation interferes with the client recognition of CK1 δ . This is certainly one explanation but another one, with a similar rationale, would be that oxidation of nearby Met side chains (for instance M405 and M414) might alter the epitope recognition of pS410 antibody. The antibody was raised against a reduced pSer410 substrate so Met side chain oxidation could lead to less binding to the blotted proteins and hence less amount of signal. Detection of PLD phosphorylation by NMR may also help here. If precipitation induced by phosphorylation is a problem, dissolving modified aggregates in urea and comparing with denatured unmodified references could also help.

The reviewer raises a very interesting point. We have now performed additional experiments to determine if methionine sulfoxidation impacts the kinetics of CK1 δ phosphorylation. As shown in **Supplementary Fig. 18**, the PLD is phosphorylated by CK1 δ even in the monomeric state (20 μ M protein, no incubation), and the phosphoPLD aggregates are dissolved in the SDS-PAGE. However, no monomeric MetO PLD is phosphorylated and only large aggregates (SDS-PAGE resistant) formed after long incubation times are phosphorylated. Therefore, methionine sulfoxidation impedes CK1 δ phosphorylation in the monomer, but not in the fibrillar aggregate. This evidence additionally validates the detection of phosphMetO PLD by the pS410 antibody.

REVIEWER COMMENTS

Reviewer #1 (Remarks to the Author):

The authors adequately addressed Reviewer 1's main concern and most minor points.

Concerning the interaction between HSP70/DNAJ/PLD (point #6), the authors justify the existence of a ternary complex in the rebuttal letter and in the manuscript at lines 333-335 page 7 by: "Evidence of interaction in additional PLD regions containing aromatic residues (e. g.: F313, F316, F367, Y374, W385, F397, F401) upon addition of HSP70" "

But the reviewer can't find more precise information about this "evidence". Does the authors mean evidence in the NMR spectra? Perturbations of the chemical shifts or lines broadening of aromatic signals? How significant is this "evidence"? Authors must specify this "evidence". Convincing experimental data using NMR and/or obtained with other methods to support the formation of a ternary complex should be presented in supplementary materials to support this conclusion. Provide that this last point is suitably addressed, the manuscript will become suitable for publication in Nature communications.

Reviewer #2 (Remarks to the Author):

The authors have made a number of revisions to the manuscript, clarifying some of the points raised in the reviews. However, some of their new data and explanations are still confusing, failing to address my original concerns regarding validity of some of the claims. Specifically:

1. As requested, the authors now show limited FRAP data. Unfortunately, there are a number of technical problems with these data, and the central claim that Met sulfoxidation impairs TDP-43 PLD LLPS remains less than convincingly documented.

(i) Liquid droplets are typically formed very rapidly, without prolonged protein incubation under LLPS-conducive conditions. Thus, it is unclear why microscopy data are shown after many hours of incubation (48 h in Fig. 1 C; 25 and 96 h in Fig S1D), during which time protein likely aggregates (see also below). Likely aggregation effects are indeed suggested by Fig. 1C (upper panel), where the features shown by DIC are largely non-spherical (as opposed of what would be expected for liquid droplets).

(ii) Given that the structures shown in the upper panel of Fig 1C are largely non-spherical, the question remains how representative is the apparently spherical droplet used for FRAP experiments in Fig. S1D. This should be documented by showing a larger field of the fluorescence micrograph that contains multiple droplets. Furthermore, it is also unclear why the 96 h FRAP data show an initial post-bleach normalized intensity around 0.4. After data processing, the initial post-bleach value should be 0.

(iii) Given (i) and (ii), it is crucial that the authors show fluorescence micrographs, FRAP data and proper phase diagrams (see below) for freshly prepared droplets.

(iv) Data shown in Fig. S1D are NOT phase diagrams (as claimed by the authors) but turbidity data as a function of time. Phase diagrams normally show under which conditions droplets are present and under which they are absent (they are typically obtained by performing experiments at different protein and salt concentrations). This issue aside, these diagrams as shown in Fig. S1D are internally inconsistent. For example, in panel A, the turbidities for 150 μ M concentration are higher than those for 300 μ M. Why? Again, this suggests that protein aggregation (which would be time-dependent) is involved here, not just LLPS.

2. Fig. 4B - Bright field light microscopy (top panel) and fluorescence microscopy (bottom panel) images under identical conditions appear to correspond. Multiple spherical species are shown in bright field microscopy image, but only one droplet is shown by fluorescence microscopy. The same problem occurs in Fig. 4F.

3. It is difficult to follow the logic why different salt conditions were used in different experiments (e.g., images shown in Fig. 1C were obtained in the presence of NaCl (presumably at 150 mM concentration, but this is even not clear from the legend), whereas "phase diagrams" in Fig. S1A (and presumably other data shown in this figure) were obtained in the presence of 10 mM KCl. Why? Some level of consistency would be required to figure out what is going on.

Reviewer #3 (Remarks to the Author):

In the revised version of the manuscript the authors included new data and figures that satisfactorily addressed all my comments.

REVIEWER COMMENTS

Reviewer #1 (Remarks to the Author):

The authors adequately addressed Reviewer 1's main concern and most minor points.

Concerning the interaction between HSP70/DNAJ/PLD (point #6), the authors justify the existence of a ternary complex in the rebuttal letter and in the manuscript at lines 333-335 page 7 by:

"Evidence of interaction in additional PLD regions containing aromatic residues (e. g.: F313, F316, F367, Y374, W385, F397, F401) upon addition of HSP70" "

But the reviewer can't find more precise information about this "evidence". Does the authors mean evidence in the NMR spectra? Perturbations of the chemical shifts or lines broadening of aromatic signals? How significant is this "evidence"? Authors must specify this "evidence". Convincing experimental data using NMR and/or obtained with other methods to support the formation of a ternary complex should be presented in supplementary materials to support this conclusion. Provide that this last point is suitably addressed, the manuscript will become suitable for publication in Nature communications.

We are very grateful to Reviewer 1 for raising this issue which was not sufficiently clarified in the previous version of the manuscript. We have now performed new experiments and generated a new Supplementary Figure (**Fig. S18** in the modified version of the manuscript) that specifically addresses this point.

Firstly, PLD crosspeaks experience signal broadening upon interaction with DNAJs since complex formation is largely on the slow exchange regime in our NMR conditions. After the addition of HSP70 to the preformed PLD:DNAJ complexes, evidence of ternary complex formation would arise from the further broadening of PLD crosspeaks. Ideally, if PLD were to interact with DNAJs and HSP70 using different regions, we would clearly observe PLD signals broadening when the ternary complex was formed. Alternatively, if HSP70 would compete with the PLD for its binding to DNAJs, we would observe several PLD crosspeaks whose Intensity would increase when we compared binary (PLD:DNAJ) and ternary complex formation. However, because the binding of the PLD to DNAJs and HSP70 largely involve identical PLD regions (see **Figs. 4C, 4G**), the differences in the PLD NMR spectra between binary and ternary complex formation are very subtle. Still, we aimed to observe differences in PLD aromatics upon formation of the PLD:HSP70:DNAJs ternary complex based on the known ability of HSP chaperones to detect bulky aromatics in IDPs (see Karagöz *et al.* Cell (2014) 156(5):963-74. doi:10.1016/j.cell.2014.01.037). Therefore, NMR signal broadening of PLD aromatics in presence of HSP70 provided evidence of ternary complex formation (see **Fig. 4G**). To clarify this point, and following the reviewer's recommendation, we have generated an additional Figure showing detailed areas of PLD's ¹⁵N HSQC spectra in presence of DNAJs and DNAJs:HSP70. Several PLD crosspeaks experience significant signal broadening in presence of DNAJs:HSP70, providing evidence of ternary complex formation. Moreover, to further validate this conclusion, we have performed additional chemical crosslinking experiments (see **Fig. S18E-F** in the modified version of the manuscript). In brief, crosslinking of the PLD:DNAJ:HSP70 ternary complex revealed high molecular weight bands in the SDS-PAGE that were specifically detected by antibodies against the three proteins, indicating that the three proteins were simultaneously crosslinked due to the formation of the ternary complex. Considering all of the above, we are confident to conclude that the ternary complex is stably formed.

Interestingly, despite the renowned power of NMR spectroscopy to detect transient biomolecular interactions, we and others have previously reported limitations in NMR spectroscopy when addressing ternary complex formation (for instance, see Jaczynska *et al.* FEBS Open Bio (2022) doi:10.1002/2211-5463.13503). In particular, in a previous study (Oroz *et al.* Nature Communications (2018) 9(1):4532. doi: 10.1038/s41467-018-06880-0) we reported the ternary complex formed by the protein tau with HSP90 and the co-chaperone FKBP51. In that study, although ternary complex formation was detected by co-immunoprecipitation and chemical crosslinking, the NMR spectra of tau remained essentially unaltered when the binary or ternary complexes were formed. The main reason for this observation was that tau's interaction with

HSP90 and KBP51 is highly polymorphic and multivalent, involving the same tau's regions in its interaction with both HSP90 and FKBP51. A similar phenomenon might explain the interaction between the PLD and DNAJs and DNAJs:HSP70.

Overall, we are grateful to the reviewer because, in the light of this issue, we were able to provide convincing data to demonstrate that the PLD forms a ternary complex with HSP70 and DNAJs, which is hindered by PLD's methionine sulfoxidation.

Reviewer #2 (Remarks to the Author):

The authors have made a number of revisions to the manuscript, clarifying some of the points raised in the reviews. However, some of their new data and explanations are still confusing, failing to address my original concerns regarding validity of some of the claims. Specifically:

1. As requested, the authors now show limited FRAP data. Unfortunately, there are a number of technical problems with these data, and the central claim that Met sulfoxidation impairs TDP-43 PLD LLPS remains less than convincingly documented.

(i) Liquid droplets are typically formed very rapidly, without prolonged protein incubation under LLPS-conducive conditions. Thus, it is unclear why microscopy data are shown after many hours of incubation (48 h in Fig. 1 C; 25 and 96 h in Fig S1D), during which time protein likely aggregates (see also below). Likely aggregation effects are indeed suggested by Fig. 1C (upper panel), where the features shown by DIC are largely non-spherical (as opposed of what would be expected for liquid droplets).

We are grateful to the reviewer for raising this concern. We agree that it was not clearly justified why the microscopy data was obtained after so prolonged incubations in the previous version of the manuscript. Precisely, the motivation was to report that, even after 96h of incubation, which was longer than the time used for the NMR relaxation experiments (Fig. S7 in the new version of the manuscript), the PLD phase-separated condensates remained fluid. In other words, we aimed to clarify that our NMR relaxation data reported on the exchange with liquid condensates rather than with solid aggregates. Still, following the reviewer's recommendations, we have performed additional FRAP data on the PLD condensates formed after 3h of incubation, showing that the recovery rates for these condensates formed after shorter incubation times are similar to those observed in the condensates formed after 24 h (see Fig. S2 in the modified version of the manuscript). We have clarified this point in the modified version of the manuscript.

In addition, because the reviewer considers that the condensates shown in Fig. 1C are "largely non-spherical", we have included new DIC and fluorescence microscopy images (see Fig. S1A in the modified manuscript) on the spherical condensates formed by the PLD after 3h of incubation at 25 °C.

Based on the data shown in Figs. 1B-C, 4A-B, 4E-F, 4H-I, S1A-D, S2, S16 we conclude that turbidity reports on liquid-liquid phase separation and not solid aggregation. Considering all of the above, we are confident that Fig. S7 reports on the exchange between disperse PLD and phase-separated fluid condensates rather than solid aggregates. Figs. 1D, S1A-B, S7 allow us to conclude that methionine sulfoxidation impedes phase separation. Following the previous study from Fawzi's group (Conicella *et al.* Structure (2016) 24(9):1537-49. doi: 10.1016/j.str.2016.07.007), where the authors provide a thorough characterization of the condensates formed by the PLD and the integral role of the PLD's double α -helix in LLPS, we conclude that the structural impact of methionine sulfoxidation on such α -helices is the basis for the reduced ability of MetO PLD to phase separate.

(ii) Given that the structures shown in the upper panel of Fig 1C are largely non-spherical, the question remains how representative is the apparently spherical droplet used for FRAP experiments in Fig. S1D. This should be documented by showing a larger field of the fluorescence micrograph that contains multiple droplets. Furthermore, it is also unclear why the 96 h FRAP

data show an initial post-bleach normalized intensity around 0.4. After data processing, the initial post-bleach value should be 0.

As mentioned above, we provide new microscopy (**Fig. S1A-B**) and FRAP (**Fig. S2**) data demonstrating the validity of the FRAP values. Regarding the normalization of the FRAP data, we have followed protocols where the post-bleach values never reached 0 intensity (corresponding to the background). Therefore, normalization covered from the maximum fluorescence intensity (in the condensate) to the minimum fluorescence intensity (in the background). Similar protocols and FRAP curves were reported in a previous study characterizing PLD condensates (Conicella *et al.* *Structure* (2016) 24(9):1537-49. doi: 10.1016/j.str.2016.07.007). This point is clarified in the modified manuscript.

(iii) Given (i) and (ii), it is crucial that the authors show fluorescence micrographs, FRAP data and proper phase diagrams (see below) for freshly prepared droplets.

As discussed above, new data on freshly prepared droplets are provided in the modified manuscript (**Figs. S1A, S2**).

(iv) Data shown in Fig. S1D are NOT phase diagrams (as claimed by the authors) but turbidity data as a function of time. Phase diagrams normally show under which conditions droplets are present and under which they are absent (they are typically obtained by performing experiments at different protein and salt concentrations). This issue aside, these diagrams as shown in Fig. S1D are internally inconsistent. For example, in panel A, the turbidities for 150 μM concentration are higher than those for 300 μM . Why? Again, this suggests that protein aggregation (which would be time-dependent) is involved here, not just LLPS.

We thank the reviewer for raising this concern. As discussed previously, we provide additional evidence in the modified manuscript supporting the conclusion that turbidity reports on LLPS and not on solid aggregation. In parallel, we have performed additional turbidity experiments to validate the ability of the PLD to phase separate. Indeed, in the new **Fig. S1C**, rather than providing additional turbidity curves, we provide a heat map representation of the concentration-dependent LLPS for PLD and MetO PLD. The intention of this particular figure is to show at which conditions the PLD forms droplets, compared to MetO PLD (by performing experiments at different protein concentrations). In particular, these experiments were performed at a fixed 150 mM KCl salt concentration, which is the salt concentration used in all the microscopy data shown in the Figures (so-called turbidity conditions), except for **Fig. 1C** and **S1B**, which contained 150 mM NaCl instead of KCl to enhance LLPS. The new **Fig. S1D** shows turbidity curves for 300 μM protein concentration and 10 mM KCl, which are the protein and salt concentrations used in the NMR experiments (so-called NMR conditions). In both experiments, using different protein and salt concentrations, we observe that methionine sulfoxidation impedes LLPS. We have clarified this point in the modified version of the manuscript.

2. Fig. 4B - Bright field light microscopy (top panel) and fluorescence microscopy (bottom panel) images under identical conditions appear to correspond. Multiple spherical species are shown in bright field microscopy image, but only one droplet is shown by fluorescence microscopy. The same problem occurs in Fig. 4F.

We thank the reviewer for raising this concern. Those images were not intended to show identical sample fields, and therefore we have reordered the figure panels to avoid possible misunderstandings. New microscopy data (new **Fig. S1A**) show simultaneous DIC and fluorescent microscopy images.

3. It is difficult to follow the logic why different salt conditions were used in different experiments (e.g., images shown in Fig. 1C were obtained in the presence of NaCl (presumably at 150 mM concentration, but this is even not clear from the legend), whereas “phase diagrams” in Fig. S1A (and presumably other data shown in this figure) were obtained in the presence of 10 mM KCl. Why? Some level of consistency would be required to figure out what is going on.

We agree with the reviewer that the experimental conditions used in the study required further

clarification. Turbidity and microscopy measurements (**Figs. 1B, 1C, 1D, 4A, 4B, 4E, 4F, 4H, 4I, S1A, S1C, S16**) were performed using 150 mM KCl. Additional measurements using 150 mM NaCl instead of KCl (**Fig. 1B, 1C, S1B**) were performed for comparison. NMR samples (**Fig. 1E-G, 4C, 4D, 4G, S1D, S3-S8**) contained 10 mM KCl. Upon LLPS, the NMR spectral quality of the PLD is significantly diminished. Therefore, we used a lower salt concentration in the NMR to improve magnet shimming to enhance the spectral quality of the long multidimensional NMR experiments in LLPS conditions. Due to this apparent discrepancy in salt concentration, we performed additional turbidity experiments using NMR conditions (300 μ M protein concentration and 10 mM KCl, **Fig. S1D** in the modified manuscript) to corroborate that the NMR measurements were obtained on the PLD undergoing LLPS. This point has been clarified in the modified version of the manuscript.

Reviewer #3 (Remarks to the Author):

In the revised version of the manuscript the authors included new data and figures that satisfactorily addressed all my comments.